# Temporal dynamics of protein complex formation and dissociation during human cytomegalovirus infection

Yutaka Hashimoto[1,2,3,4], Xinlei Sheng[1], Laura A. Murray-Nerger[1] & Ileana M. Cristea[1]*

The co-evolution and co-existence of viral pathogens with their hosts for millions of years is reflected in dynamic virus-host protein-protein interactions (PPIs) that are intrinsic to the spread of infections. Here, we investigate the system-wide dynamics of protein complexes throughout infection with the herpesvirus, human cytomegalovirus (HCMV). Integrating thermal shift assays and mass spectrometry quantification with virology and microscopy, we monitor the temporal formation and dissociation of hundreds of functional protein complexes and the dynamics of host-host, virus-host, and virus-virus PPIs. We establish pro-viral roles for cellular protein complexes and translocating proteins. We show the HCMV receptor integrin beta 1 dissociates from extracellular matrix proteins, becoming internalized with CD63, which is necessary for virus production. Moreover, this approach facilitates characterization of essential viral proteins, such as pUL52. This study of temporal protein complex dynamics provides insights into mechanisms of HCMV infection and a resource for biological and therapeutic studies.

[1] Department of Molecular Biology, Princeton University, Lewis Thomas Laboratory, Washington Road, Princeton, NJ 08544, USA. [2] Division of Morphogenesis, Department of Developmental Biology, National Institute for Basic Biology, Okazaki, Aichi 444-8585, Japan. [3] International Research Collaboration Center, National Institutes of Natural Sciences, Tokyo, Japan. [4] Department of Cell Biology, Nagoya City University Graduate School of Medicine Sciences, Nagoya, Aichi 467-8601, Japan. *email: icristea@princeton.edu

Virus replication within host cells proceeds with exquisite spatial and temporal control. To accomplish this, stereotyped biochemical interfaces between virus and host components, such as those mediated by protein–protein interactions (PPIs), are essential. PPIs provide the building blocks for macromolecular complexes, which are dynamically regulated throughout viral replication. Across diverse virus families, cellular protein complexes are co-opted for virus entry into host cells, virus-stimulated transcription and translation, organelle remodeling, and assembly and egress of infectious viral particles. This virus–host interplay is evident for the clinically relevant pathogen, human cytomegalovirus (HCMV). This widespread β-herpesvirus establishes lifelong, usually latent and asymptomatic infections[1]. However, HCMV is associated with morbidity and mortality in immunocompromised individuals[2] and a leading cause of virally induced birth defects[3]. Despite its impact on human health, currently there is no vaccine, and anti-viral therapies are limited by toxicity and drug resistance.

Being the largest known human herpesvirus (encoding >200 conventional reading frames)[4], HCMV has the means to regulate numerous PPIs and cellular protein complexes to ensure its replication and propagation[5,6]. Initiation of infection depends on viral glycoprotein complexes (gH/gL, gB) that bind cell-specific membrane receptors, including heparan sulfate glycosaminoglycans, epidermal growth factor receptor, platelet-derived growth factor receptor-α and integrins, leading to cell entry via endocytic internalization pathways[7]. The intracellular environment is then tailored to promote HCMV replication[8], partly through stimulation of messenger RNA (mRNA) transcription and translation of viral products via mammalian target of rapamycin (mTOR) signaling and increased assembly of the eIF4F cap recognition complex[9]. For example, the anti-apoptotic viral protein pUL38 interacts with the tuberous sclerosis tumor suppressor protein complex (TSC1/2) to maintain an active mTOR pathway[10]. To facilitate HCMV assembly and egress, host and viral proteins are temporally coordinated between organelles to form the cytoplasmic virion assembly complex[8,11,12].

Dynamic regulation of PPIs also underlies host defense and virus immune evasion mechanisms[13]. Members of the NF-κB/IκB complex are dissociated upon infection with various viruses, leading to the transactivation of NF-κB responsive genes that act in immune response[14]. Similar to many viruses, HCMV has acquired mechanisms to prevent IFN responses and apoptosis[15]. For instance, the HCMV immediate-early protein IE1 inhibits the functions of cellular ND10/PML nuclear bodies, which normally induce immune signaling and suppress virus gene expression[16]. Therefore, understanding PPI dynamics not only provides insights into virus biology, but can also uncover antiviral targets.

To date, virus–virus and virus–host PPIs have been largely studied using yeast two-hybrid approaches[17,18], quantitative mass spectrometry (MS)[19], and immunoaffinity purification (IP)[20]. We and others have developed approaches that paired IP with quantitative MS to identify interactions during the progression of an infection[20]. Altogether with structural and functional studies, PPI repositories have been generated[21–23], providing datasets for predicting host and virus PPIs[24] and for uncovering virally targeted complexes[25,26]. Yet, immunoprecipitation-mass spectrometry (IP-MS) has limited scalability, as each bait requires a distinct biochemical isolation and, frequently, molecular cloning. Other challenges include limited availability of IP-grade antibodies, compromised protein functions by epitope tagging, and difficulty in assigning protein complex membership. While protein correlation profiling-MS can inform on complex composition[27,28], its analytical scale and resolution limits throughput and proteome coverage. This emphasizes the need for complementary approaches that monitor protein complex

dynamics at a systems-level. Towards this goal, recent work used thermal proteome profiling[29] (TPP) and thermal proximity coaggregation (TPCA) paired with MS quantification (MS-CETSA) to monitor proteome-wide protein complex dynamics in cells and tissues[30]. This method uses the principle that proteins within a complex co-aggregate upon heat denaturation. Thereby, protein complex components would exhibit similar, non-random progressive insolubilities when subjected to an increasing temperature gradient. The soluble proteome fractions are quantified using multiplexed MS as a function of temperature, allowing high-throughput construction of thermal denaturation curves. MS-CETSA was initially applied to studying drug targets in cells and tissues[29,31] and recently to monitoring protein complex dynamics during the cell cycle[32]. However, no study has yet used TPP or MS-CETSA in infected cells.

Here, we investigate system-wide protein complex dynamics during the HCMV replication cycle. We monitor the formation and dissociation of hundreds of functional protein complexes, as well as the dynamics of host–host, virus–host, and virus–virus PPIs during infection. With validation, microscopy and functional assays, this thermal profiling analysis allowed us to discover host complexes and protein co-translocations necessary for virus production, as well as interactions with immune factors for an essential viral protein. Altogether, this study provides insights into mechanisms underlying HCMV infection and a resource for future studies focused on HCMV biology and pathogenesis or the discovery of therapeutic targets.

## Results

**TPCA analysis during HCMV infection.** To investigate the impact of HCMV infection on the formation and dissociation of protein interactions, we used the concept of TPP and performed a TPCA analysis (Fig. 1a)[30]. Primary human fibroblasts were collected as uninfected (mock) or following infection, at time points representing different HCMV replication cycle stages, i.e., early (24 h post infection (hpi)), delayed early (48 hpi) and late (72 hpi and 96 hpi). At each time point, the cells were subjected to stepwise increases in temperature (37 to 64 °C). Soluble proteins at each temperature were quantified by labeling with 10-plex tandem mass tags (TMT) and MS. Approximately 5300 cellular and viral proteins were identified across biological replicates (Supplementary Fig. 1a and Supplementary Data 1), and the correlation coefficients indicated reproducibility when data normalization was performed either by reporter ion intensity or normalized adjusted solubility (Fig. 1b and Supplementary Fig. 1b, e). Viral protein abundances increased during infection, in agreement with progression through viral replication (Fig. 1c and Supplementary Fig. 1c). Plotting of proteome abundances in two-dimensional principal component analysis (PCA) space showed separation of the data both with temperature increases (Supplementary Fig. 1d) and infection stages (Fig. 1d). This separation suggests that protein abundance changes and aggregation curves contain information regarding alterations occurring during infection. A range of factors, not limited to PPIs, can drive these alterations, including associations with small molecules or nucleic acids, changes in post-translational modifications (PTMs) or subcellular localization[33–35]. Given this diversity of contributing factors, we first focused on members of known functional cellular protein complexes. We then expanded this analysis to predict the formation and dissociation of host–host, virus–host, and virus–virus PPIs.

**HCMV induces temporal alterations in protein complexes.** To assess the ability of HCMV to modulate human cellular protein complexes, we used information from the CORUM database[36] (Fig. 1e) and determined whether proteins quantified in our

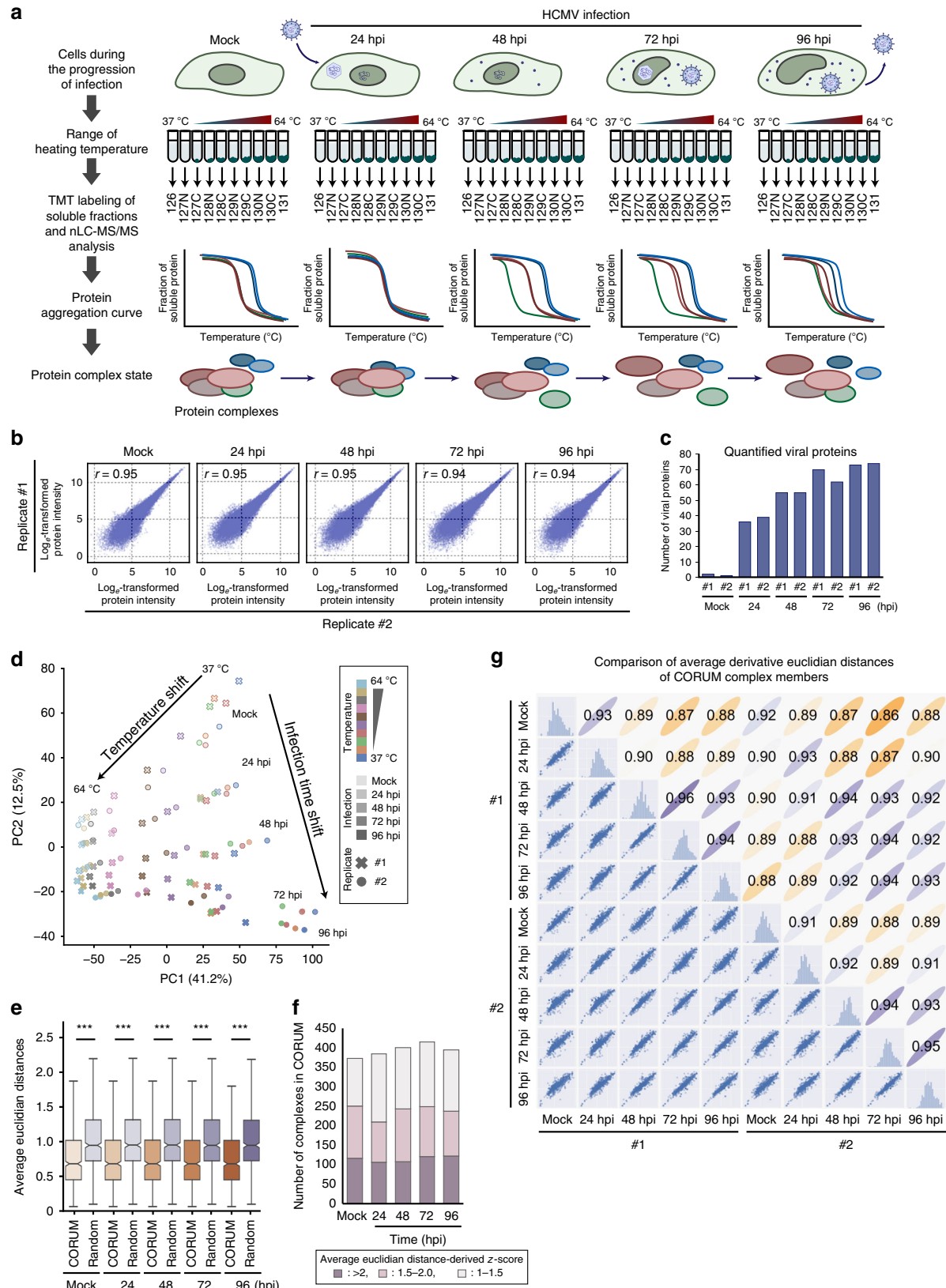

datasets are known to be present within certain complexes (Supplementary Fig. 2). Average Euclidean distances were calculated between protein aggregation curves of complex members to predict associations or dissociations. As expected, average Euclidean distances were smaller for complex members than for random non-complexed proteins at all time points. Over 200

complexes were monitored with high confidence ($z$-score >1.5) (Fig. 1f), and reproducible identification was reflected in the >0.93 correlation coefficient of the average Euclidean distance for each complex between replicates (Fig. 1g). Correlation coefficients gradually decreased from uninfected to late infection, suggesting the modulation of complexes during infection (Fig. 1f).

**Fig. 1 MS-CETSA workflow for uncovering the association and dissociation of protein complexes and virus–host protein interactions during HCMV infection. a** MS-CETSA workflow for sample collection, TMT labeling, and data analysis during HCMV infection. **b** Correlation between replicates (Pearson's r-value) at each infection time point, calculated as $\log_e$ transformed protein abundances that were estimated from the sum of the peptide reporter ion intensities for peptides shared across time points for the indicated proteins. **c** Number of quantified viral proteins in each replicate. **d** PCA plot of all protein abundances showing the separation of TMT channels (temperature) and HCMV time points. Protein abundance was calculated as the sum of all peptide reporter ion intensities for each protein. Data from each replicate is indicated by the shape of the marker. **e** Comparison of average Euclidean distances between protein complexes from the CORUM database or random selections of proteins. A Mann–Whitney U-test was performed, ***p-value < 0.001. **f** The number of identified protein complexes documented by the CORUM database with different z-score thresholds (1, 1.5, and 2). z-score describes the number of standard deviations of the CORUM complex proteins from the mean of the randomly selected proteins. High z-scores represent high Ex and low-average Euclidean distance, which indicates closer melting curves of the complex components. **g** Scatter plot matrix comparing the average Euclidean distances of protein complexes documented by CORUM between samples from both replicates. Each dot in the lower triangle represents the derivative of average Euclidean distance of one protein complex (Ex). Pearson's r-values are shown for each pair of samples in the top triangle.

Temporal changes in protein complexes were clustered by Fuzzy c-means, and GO enrichment analysis for each cluster indicated biological functions associated with complexes predicted as stabilized or disrupted (Fig. 2a, Supplementary Data 2, and Supplementary Fig. 3). Giving confidence to this analysis, several modulated functions and pathways were expected during HCMV infection. For example, cluster #8 with protein complexes gradually destabilized during infection contained mitochondrial and cell adhesion complexes. This observation aligns with the known virus-induced fragmentation of mitochondria to suppress apoptosis and modulation of cellular metabolism[37,38]. Complexes stabilized after 48 hpi (cluster #4) included those related to translation initiation (Fig. 2b), consistent with the HCMV-induced eIF4F assembly and cap-dependent translation[39,40].

Protein complexes involved in cytoskeleton organization and transmembrane receptor protein tyrosine kinase signaling pathway appeared stabilized from 24 to 96 hpi (cluster #3). The latter category included complexes related to the phosphoinositide 3-kinase (PI3K) pathway, which displayed increased stability at 24 and 48 hpi (Fig. 2c). PI3K signaling is known to be activated following HCMV entry into host cells[41,42]. Indeed, our interaction network representation of PI3K pathway related proteins showed reduced Euclidean distances at 24 hpi, especially between PI3K regulatory subunit alpha (PIK3R1 or p85α) and its interacting proteins (Fig. 2d and Supplementary Fig. 4a).

Our analysis also pointed to infection-induced regulation of complexes involved in protein transport (Fig. 2a, cluster#1). Predicted as stabilized were the Wiskott–Aldrich syndrome protein and SCAR homolog (WASH) and the COMMD/CCDC22/CCDC93 (CCC) complexes, which are critical for cargo sorting and endosomal trafficking[43,44] (Fig. 3a). The CCC complex is formed of copper metabolism MURR1 domain-containing proteins (COMMD1-10), C16orf62, and the coiled-coil domain-containing proteins 22 and 93 (CCDC22 and CCDC93)[43,44] (Fig. 3b). The CCC and WASH complexes have been reported to interact to sort specific cargos, such as the low-density lipoprotein receptor (LDLR)[44] and the copper transporter ATP7A[43]. The regulation of endosomal trafficking is critical during HCMV infection, including for HCMV assembly complex formation and virion egress[45]. However, the roles of the CCC and WASH complexes in these processes remain unknown. Euclidean distances were reduced between WASHC2C, a major component of the WASH complex, and the CCC complex at 48 and, in particular, at 72 hpi compared to mock (Fig. 3a, b and Supplementary Fig. 4b). This finding suggests increased CCC-WASH association during infection. To validate this, we generated and transfected FLAG-tagged WASHC2C into fibroblasts, performed IPs in uninfected and infected cells, and quantified WASHC2C association with CCC members using targeted MS (parallel reaction monitoring, PRM). Indeed, at 72 hpi, the interaction between FLAG-WASHC2C and CCC components (C16orf62, CCDC22, and CCDC93) was increased

(Fig. 3c). To assess whether CCC-WASH may support virus production or host defense, we performed small-interfering RNA (siRNA)-mediated WASHC2C knockdowns and confirmed knockdown maintenance throughout infection and cell viability (Fig. 3d, e and Supplementary Fig. 5a). WASHC2C knockdown resulted in significantly reduced virus titers (Fig. 3f). Our association data suggests that WASHC2C is modulated late in infection. To support this, we performed western blot analysis of immediate early (IE1), delayed early (pUL26), and late (pUL99) viral proteins. IE1 levels were similar in siWASHC2C and siControl cells at 24 hpi (Fig. 3g), indicating WASHC2C does not affect viral entry. pUL26 and pUL99 levels were not decreased, suggesting WASHC2C has a role late in infection (after viral protein production), likely in intracellular trafficking events. The slightly increased levels of pUL26 and pUL99 at 96 and 120 hpi in siWASHC2C cells may indicate impaired trafficking of viral particles during assembly or egress. As the cellular sorting machinery and endocytic trafficking pathways are modulated for efficient virus production, our results uncover a previously unrecognized regulatory point late in HCMV infection via the CCC-WASH association (Fig. 3h).

**The HCMV receptor integrin beta 1 is internalized with CD63**. In addition to monitoring members of protein complexes, we sought to understand the regulation of individual proteins that may play roles during infection, while not necessarily being known components of macromolecular complexes. To accomplish this, we measured melting temperatures (Tm) of individual proteins using the MS-cellular thermal shift assay (MS-CETSA)[31]. A change in protein associations, localization, structure or function could lead to a change in its thermal stability[33–35]. Therefore, we calculated the absolute ΔTm values (|ΔTm|) for each protein from uninfected cells to each infection time point (24, 48, 72, 96 hpi) (Fig. 4a and Supplementary Data 3). We considered the possibility that infection-induced changes in protein abundances can contribute to Tm changes. However, changes in protein abundances had low correlations (Pearson's r < 0.1) with ΔTm for every infection time point suggesting that, for most proteins, alterations in abundances are not the primary drivers of Tm changes (Supplementary Fig. 6a). Furthermore, enrichment analysis of curated gene sets from MsigDB[46] showed that HCMV infection related gene sets[47] displayed altered |ΔTm| (Fig. 4b), suggesting this measurement captures infection-induced temporal changes in protein properties. Global enrichment analysis using the KEGG pathway database showed that one of the most enriched terms was ECM_RECEPTOR_INTERACTION (Fig. 4c). Among this subset were proteins critical for cell-cell adhesion and cell-extracellular matrix (ECM) interactions, including integrin beta 1 (ITGB1), a receptor for HCMV entry[48]. To understand what may drive this change in ITGB1 Tm during infection, we monitored the Tm values of functionally related complexes and

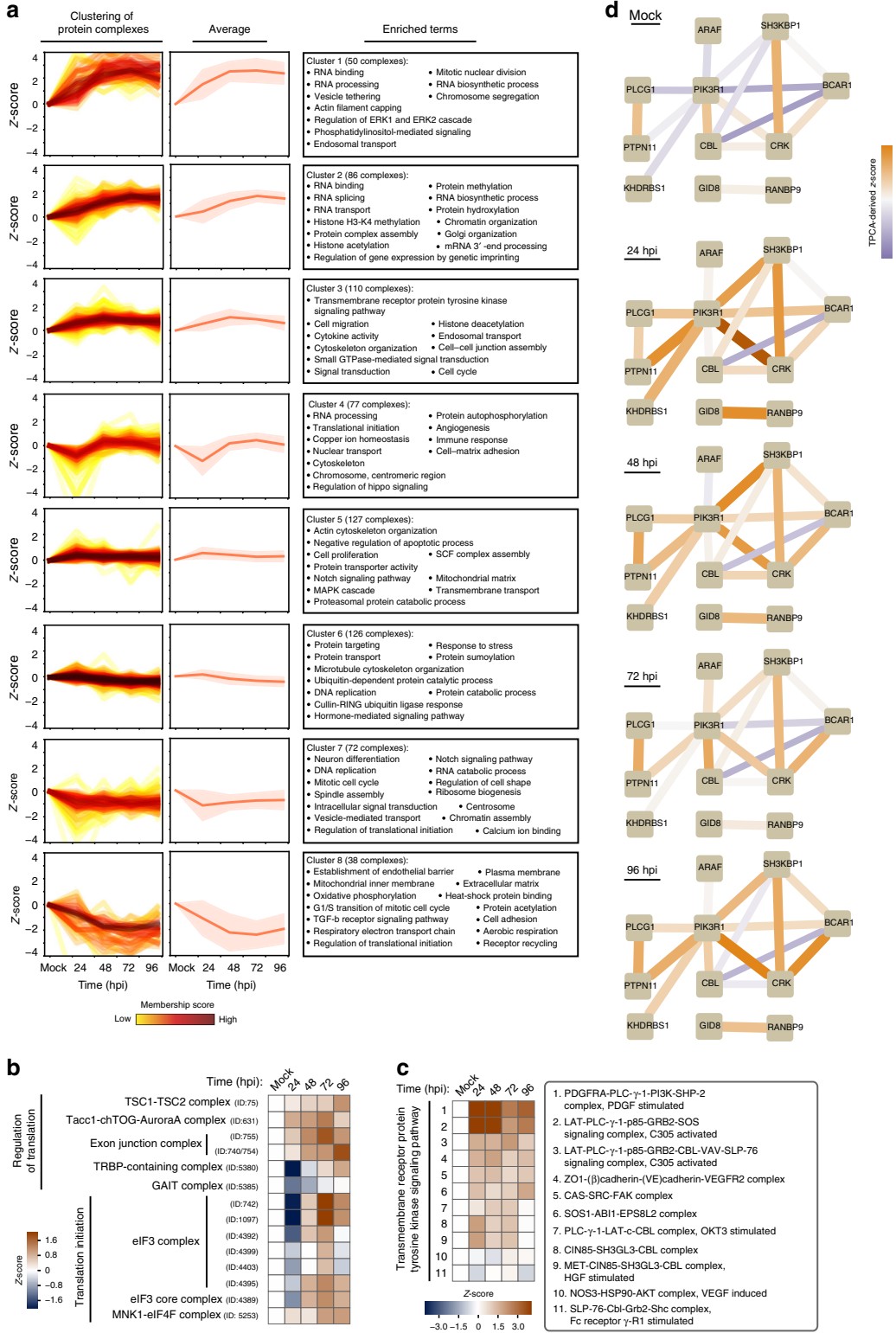

**Fig. 2 Temporal changes in functional protein complexes point to the stabilization of translation and PI3K signaling pathway complexes during HCMV infection. a** (Left) Euclidean distance-based z-score plots shift from mock after infection for CORUM protein complexes. The corresponding z-score represents the number of standard deviations between the average Ex and the mean of the fitted Gaussian distribution. Protein complexes are clustered by Fuzzy c-means and membership scores are depicted by the color of lines. Membership score represents how close to the center of the cluster a given complex is. Higher membership scores (dark red line) indicate how close the given protein complex is to the average of its corresponding cluster, while lower membership scores indicate larger deviation of the complex from this average. (Middle) Average z-score plot for each cluster (solid line) ±SD (shaded region). (Right) GO terms enriched in each cluster. **b** Heat map of z-scores for translation-related CORUM protein complexes. **c** Heat map of z-scores for CORUM protein complexes regulating transmembrane receptor protein tyrosine kinase signaling. **d** Network plots of the proteins in the PI3K pathway throughout infection. Edge width and color represent transformed Euclidean distance between nodes and the corresponding z-score derived from TPCA, respectively.

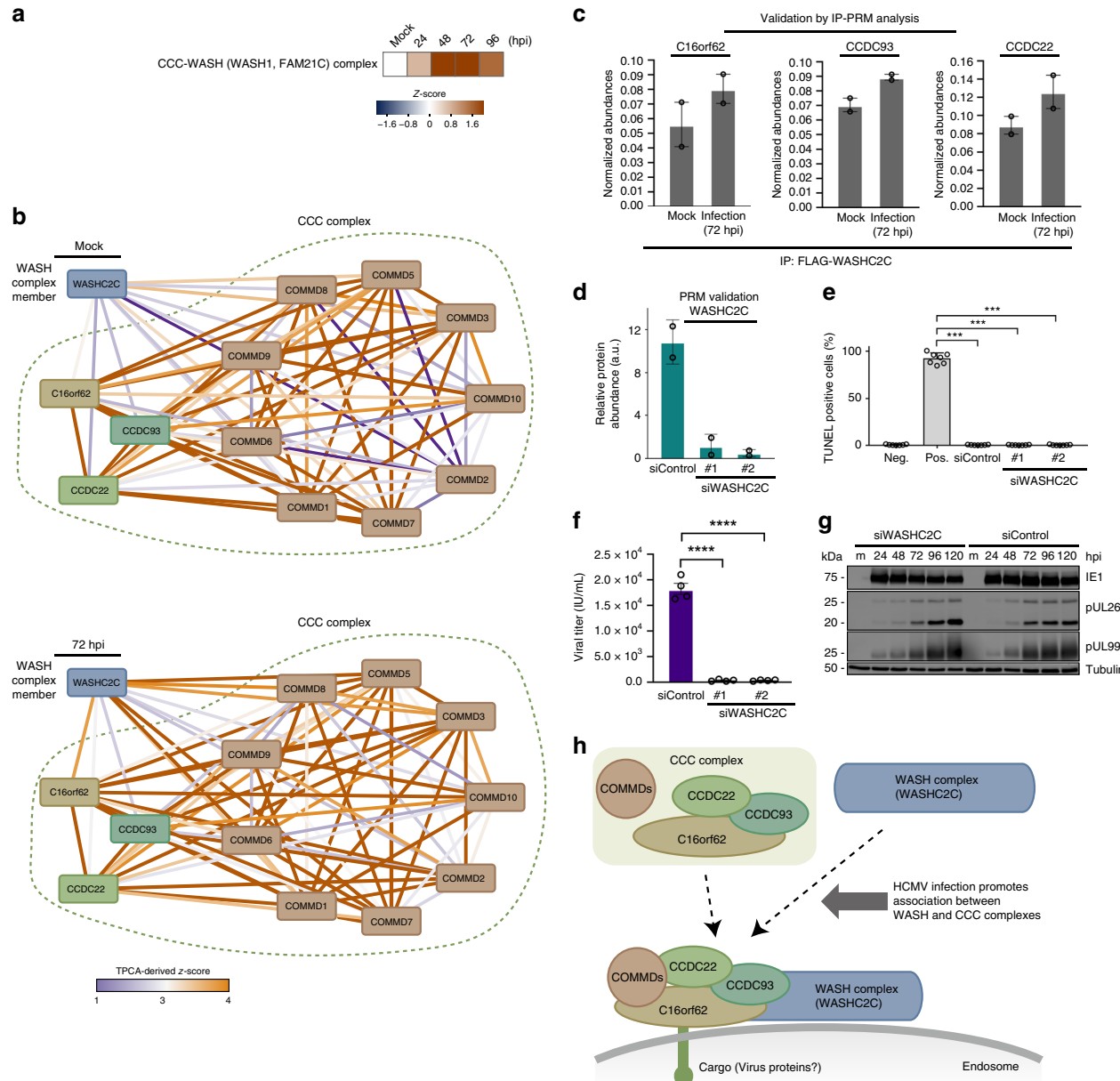

**Fig. 3 HCMV infection enhances the stability of the CCC-WASH protein complex that is needed for virus production. a** Heat map of $z$-scores for the CCC-WASH complex at each infection time point. **b** Network plot of proteins involved in CCC-WASH complexes in mock and infected (72 hpi) cells. Edge width and color indicate transformed Euclidean distance between nodes and their $z$-score, respectively. Low $z$-score suggests a weak association (lines with blue tones), while a high $z$-score suggests a stronger association (lines with orange tones) between proteins. **c** Validation of the stabilization of the CCC-WASH complex by IP-PRM analysis. FLAG-tagged WASHC2C IPs were performed in mock and infected (72 hpi) cells, and the normalized protein abundances were determined by PRM for C16orf62, CCDC93, and CCDC22. Data are mean ± SEM ($n = 2$). **d** PRM validation of WASHC2C siRNA-mediated knockdown. Data are mean ± SD ($n = 2$). **e** TUNEL assay for cells transfected with siRNAs targeting WASHC2C or siControl, $n \geq 6$, ***$p$-value < 0.001. **f** Viral titers from cells transfected with siRNAs targeting WASHC2C or a non-targeting siRNA control. Data are mean ± SEM and a two-sided Student's $t$-test was performed, $n = 4$ biological replicates, ****$p$-value < 0.0001. **g** Western blot showing viral protein levels for each of the three temporal classes for siWASHC2C cells or siControl cells. Source data are provided as a Source Data file. **h** Proposed model for the enhanced association between the CCC complex and the WASH complex during HCMV infection. HCMV infection promotes the stabilization of the CCC complex and the gain of interaction between the CCC and WASH complexes on the endosome membrane to support the intracellular transport of cargos.

proteins from CORUM (Fig. 4d and Supplementary Fig. 4c). The Euclidean distances between protein aggregation profiles indicated that, while ITGB1 and collagen (COL1A1) or fibronectin (FN1) became gradually distant during infection, ITGB1 and CD63 became closer. CD63 activates ITGB1 and promotes integrin signaling[49], as well as enhances the internalization of several plasma membrane proteins[49,50]. Our results suggest that CD63 may similarly promote ITGB1 internalization during

infection, but this was not previously demonstrated. To test this hypothesis, we assessed ITGB1 and CD63 localizations by immunofluorescence microscopy. Indeed, ITGB1 had a pronounced intracellular localization at 96 hpi compared to uninfected cells, displaying some co-localization with CD63 (Fig. 4e). Additionally, ITGB1 levels decreased during infection (Fig. 4f). To determine the role of CD63 in ITGB1 internalization and virus production, we performed siRNA-mediated CD63

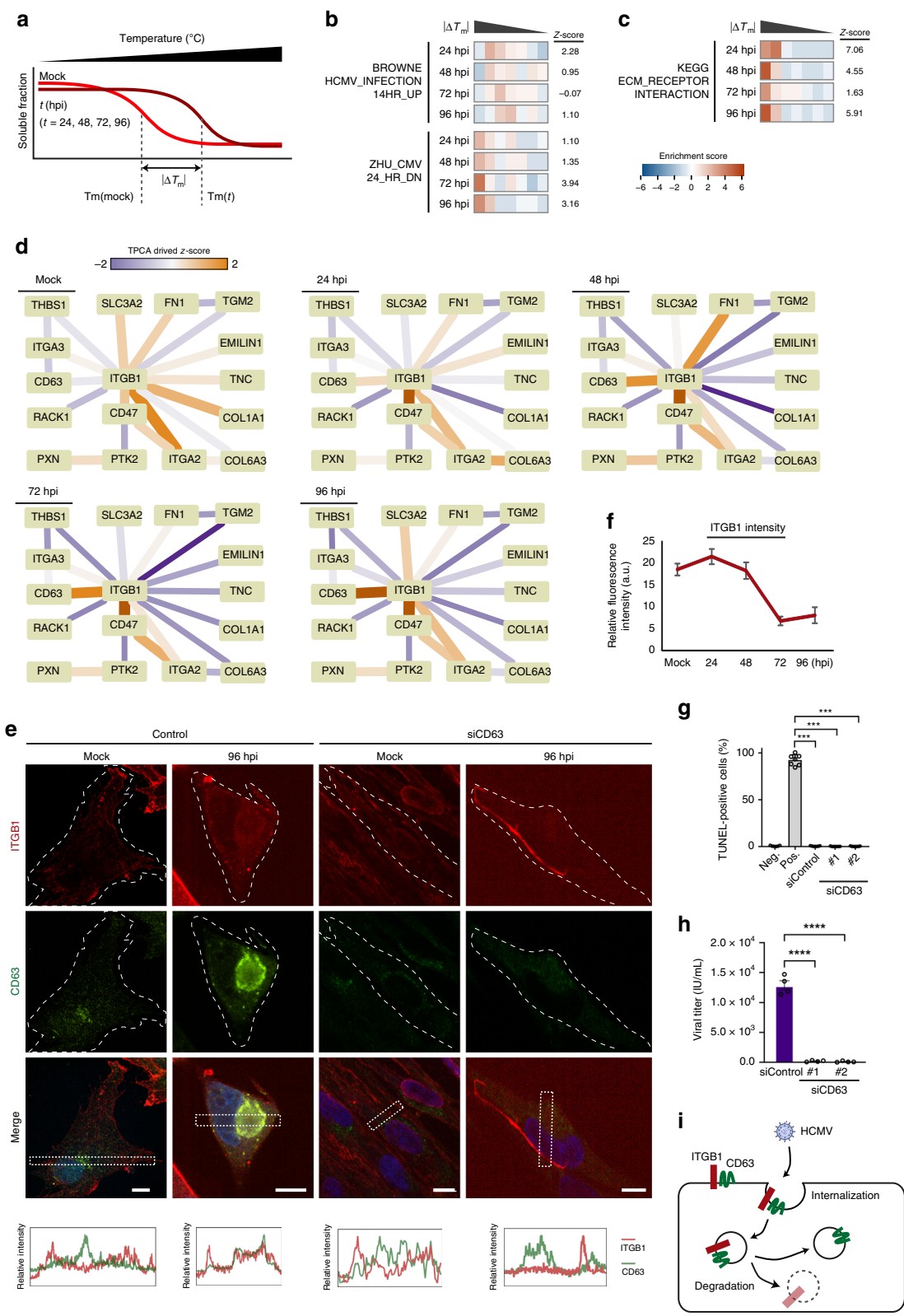

knockdowns and confirmed knockdown throughout infection (Supplementary Fig. 5b) and cell viability (Fig. 4g). By microscopy, we observed more ITGB1 retained at the plasma membrane at 96 hpi upon CD63 knockdown in comparison to control (Fig. 4e). Moreover, we observed significant reductions in virus titers upon CD63 knockdown (Fig. 4h). Altogether, our results demonstrate that, prior to its degradation, ITGB1 internalization is in part mediated via association with CD63, with CD63 exhibiting a pro-viral function (Fig. 4i).

**Infection alters fatty acid metabolism in part via ACAT1.** Although our analysis based on |ΔTm| captured numerous protein alterations, it missed the subset of proteins that display non-

**Fig. 4 Integrin beta 1 (ITGB1) develops an association and is internalized with the pro-viral tetraspanin protein CD63. a** Schematic view of how absolute ΔTm is calculated. The inflection point of each aggregation curve is defined as the Tm. The absolute value of the difference in Tm difference between an infection time point and mock is the absolute ΔTm. **b** Enrichment analyses of gene sets related to HCMV infection based on |ΔTm|. Gene sets known to be regulated by HCMV infection were acquired from MsigDB. Proteins in these gene sets were divided evenly into seven bins by |ΔTm|, and their |ΔTm| values were subjected to enrichment analysis using information theory. **c** Enrichment analysis of the ECM_RECEPTOR_INTERACTION gene set from the KEGG database. **d** Network plots of the integrin β1 related protein complex constructed from CORUM database during infection. Changes in interactions are shown over the course of HCMV infection. **e** Immunofluorescence analysis of ITGB1 and CD63 in uninfected (mock) and infected cells (96 hpi) in control and siRNA-mediated knockdown of CD63 (siCD63) cells. Fluorescence intensities for ITGB1 and CD63 across the sections are plotted below. Scale bar 10 μm, n = 9 cells. **f** Average fluorescence intensity within a ROI ± SD for ITGB1 during infection, mock and 24 hpi, n = 13; 48 hpi, n = 10; 72 hpi, n = 11; 96 hpi, n = 9. **g** TUNEL assay for cells transfected with siRNAs targeting CD63 or siControl, n = 6, **p-value < 0.001 **h** Viral titers ± SEM for siCD63 cells or siControl cells, n = 4, ****p-value < 0.0001. **i** Proposed model for ITGB1 sequestration by CD63 at the plasma membrane. Upon HCMV infection, CD63 mediates ITGB1 internalization, which leads to ITGB1 degradation.

sigmoidal melting curves. To overcome this limitation, we monitored the difference in temporal Euclidean distance (TED) (Fig. 5a, Supplementary Fig. 6b, and Supplementary Data 3). This measurement assessed the change in protein solubility across infection time points, providing analogous information to absolute area under the curve (Supplementary Fig. 7). TED had an overall positive correlation with |ΔTm| (Supplementary Fig. 8) and, as proof of concept, we verified that TED recapitulated our |ΔTm| results for ECM receptor interaction proteins (Fig. 5b). Enrichment analysis of TED measurements using the Hallmark gene set[46] showed an enrichment in mitochondrial oxidative phosphorylation. We confirmed this enrichment by analyzing two additional mitochondrial gene sets (Mootha[51] and Wong[52], Fig. 5c), and noted that this enrichment would not have been detected using |ΔTm| measurements, highlighting the need for this complementary analysis (Supplementary Fig. 9a). Using TED and measuring normalized enrichment scores (NES), we observed that genes related to mitochondrial metabolism displayed some of the most significant temporal changes (Fig. 5d, Supplementary Data 4, and Supplementary Fig. 9b). This agrees with prior knowledge that HCMV infection rewires mitochondrial metabolism-related proteins during lytic infection.

Upregulation of fatty acid synthesis is necessary for generating lipids that are incorporated into the virus envelope and producing infectious virus particles[53]. Alterations in fatty acid metabolism proteins were evident from our measurement of the derivative of Euclidean distances (Ex = 1/(1 + E)). In contrast to the Ex values for all proteins in the cellular proteome, which remained relatively unchanged over time, proteins from the fatty acid metabolism pathway displayed a time-dependent shift in Ex values (Fig. 5e). This shift could represent changes in protein associations with other proteins, metabolites or lipids[33–35]. A closer look at individual proteins in the fatty acid metabolism pathway (Fig. 5f) showed that acetyl-CoA acetyltransferase (ACAT1) exhibited increased TED values as the infection progressed. ACAT1 catalyzes the last step of fatty acid beta-oxidation in the mitochondria, and its levels are induced by HCMV[54], although its role during infection has not been yet investigated. We propose that ACAT1 can act to restrict the rate of lipogenesis during infection. Indeed, ACAT1 siRNA-mediated knockdowns elevated virus titers (Fig. 5g, h), suggesting ACAT1 would normally restrict virus production.

**IGF2R exhibits dynamic distribution and is proviral.** Previous work has established that HCMV induces substantial organelle remodeling and protein translocations between organelles[12], and protein localization changes can affect protein Tm[34,35]. Therefore, we interrogated our data to determine if proteins known or predicted to translocate display changes in aggregation properties. We assessed two resources of translocating proteins—the translocatome database[55] (translocations outside the context of infection) and our report of potentially translocating proteins during HCMV infection[12] (Supplementary Data 5). Surprisingly, for both datasets, most Tm values were not changed at any infection time point compared to mock (Fig. 6a and Supplementary Fig. 10a). When measuring the derivative of TED, a subset of proteins had changes in thermal stability, and this was mainly attributed to ribosomal proteins (Fig. 6b, orange; and Supplementary Fig. 10b). Most remaining proteins had similar thermal stabilities (Fig. 6b, green), although the differences were statistically significant. ΔTm and TED values were weakly positively correlated, leading to similar conclusions (Fig. 6a and Supplementary Fig. 10a). This comparison further supported that TED may better capture shifted thermal profiles that cannot be fitted to sigmoidal curves. This is the case for proteins with Tm values >65 °C (data points at −10 log_eΔTm). Altogether, these analyses suggest that protein subcellular localization changes do not always lead to altered protein aggregation properties. It is possible that many translocating proteins move together with their other complex components. However, some translocating proteins with altered Tm values were observed. This included expected changes, such as for beta-2-microglobulin (B2M) and transitional endoplasmic reticulum ATPase (VCP), both of which are modulated by HCMV[56,57] (Fig. 6a).

Altered Tm values were also observed for the insulin-like growth factor type II receptor (IGF2R), a protein reported to localize at the virus assembly complex[58], but not yet functionally studied during infection. Using microscopy, we confirmed IGF2R localization to the assembly complex (marked by pUL99) at 96 hpi (Fig. 6c). As IGF2R had altered Tm values at all infection time points (Fig. 6a), we monitored its localization throughout infection (Fig. 6d). While IGF2R primarily localized to the perinuclear region in uninfected cells, early in infection (24 hpi) it developed a less polarized phenotype (Fig. 6e). At 48 hpi, IGF2R displayed prominent localization around the nucleus, which by 72 hpi concentrated to the viral assembly complex (Fig. 6d). Altogether with our Tm data, these results suggest the regulation of IGF2R at both early and late stages of infection (Fig. 6f). To determine whether IGF2R is needed for viral replication, we generated and validated IGF2R CRISPR knockout (KO) cells (Fig. 6g), which did not impact cell viability (Fig. 6h). Viral titers were decreased in CRISPR KO cells compared to control cells (Fig. 6i), suggesting that IGF2R serves a pro-viral function, possibly by regulating viral protein transport.

**Viral and host proteins have distinct aggregation properties.** Having applied this approach to investigating cellular protein complexes, we next assessed viral proteins. On average, viral proteins displayed lower melting temperatures than cellular proteins (Fig. 7a). One explanation is that proteins with higher molecular weights tend to have lower Tm[34,59]. Indeed, the viral proteins we identified in this dataset have higher molecular

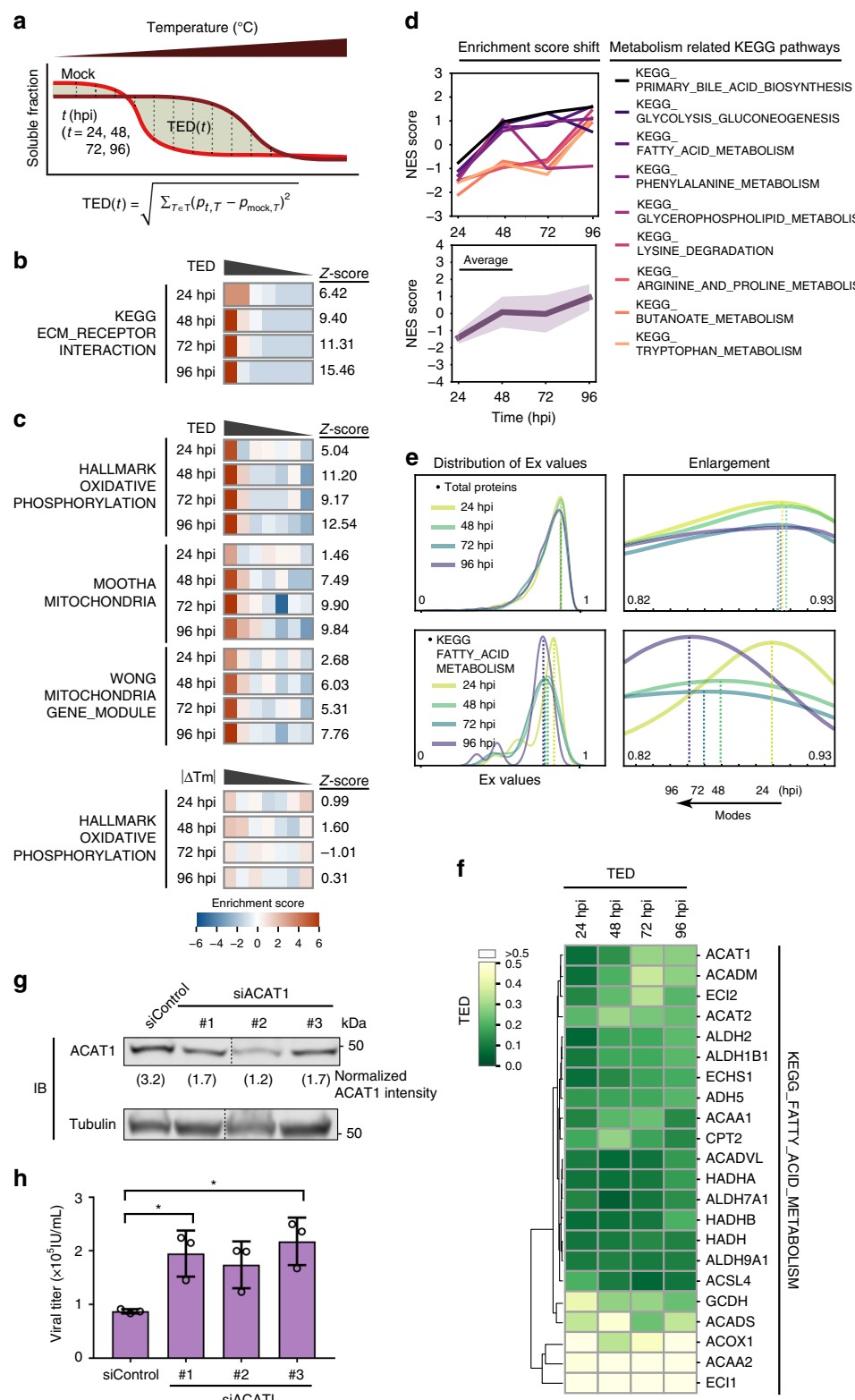

weights on average than cellular proteins (Fig. 7b). Additionally, a protein Tm is affected by its hydrophobicity and charged vs. polar non-charged residues (CvP)[60]. Tm negatively correlated with molecular weight for both cellular and viral proteins (Fig. 7c). However, this correlation was not observed for grand average of hydrophobicity (GRAVY) or CvP bias (Supplementary Fig. 11a). Therefore, differences in molecular weight partly drive differences in Tm values between viral and host proteins. Other factors, such

as the structures of host and viral proteins, may further contribute to Tm value differences. We next investigated viral proteins based on their localizations to virion compartments, i.e., tegument, envelope and capsid (Fig. 7d and Supplementary Fig. 11b). Tegument proteins, which are part of incoming infectious particles, showed high variability in their coefficients of variation of Ex, especially when comparing early time points (Supplementary Fig. 11b). This variability aligns with their dynamic functions

**Fig. 5 HCMV infection induces variability in proteins involved in fatty acid metabolism, including the antiviral factor ACAT1. a** Schematic of how TED is calculated. The aggregation curves of one protein at mock and the indicated infection time ($i$) are plotted, and the area between the two curves is TED($t$). $\text{TED}(t) = \sqrt{\sum_{T \in \mathbb{T}} \left( p_{t,T} - p_{mock,T} \right)^2}$. **b** Enrichment analysis based on TED of the ECM_RECEPTOR_INTERACTION gene set from the KEGG database. **c** Heat maps of enrichment analysis using TED for Hallmark oxidative phosphorylation and other mitochondrial related gene sets (Mootha and Wong). Enrichment analysis for the Hallmark oxidative phosphorylation gene set using ΔTm is shown below for comparison. **d** NES score graphs for metabolism-related KEGG pathways. Metabolic pathways that show the highest NES shifts between late and immediate-early stages of infection were selected. The individual NES scores for each pathway (top) and the averaged NES scores (bottom) are displayed. Shading in the bottom plot delineates SD. **e** The distribution of derived Euclidean distances (Ex) for all proteins and for the fatty acid metabolism pathway. Kernel density estimation plot of Ex for all proteins (top) and proteins involved in FATTY ACID METABOLISM (from KEGG pathway) (bottom) are shown with enlarged snapshots of the peak area (right). **f** Hierarchical clustering of proteins in the fatty acid metabolism pathway by Ward's method. Heat map color indicates TED values. **g** Western blot confirmation of siRNA-mediated ACAT1 knockdown with three siRNA constructs. Tubulin was used as the loading control, and the normalized ACAT1 intensities based on densitometry are shown below the ACAT1 blot. Source data are provided as a Source Data file. **h** Viral titers ± SEM from cells transfected with ACAT1 siRNAs or control siRNA, $n = 3$, *$p$-value < 0.05.

early in infection, such as inhibition of apoptosis and immune signaling. The higher variability observed for other virion components likely reflects changing properties during the viral assembly process. In contrast, viral proteins produced during infection but not incorporated in virus particles had lower variability. These cellular viral proteins may stay within certain complexes during infection.

To better understand this distinction between viral protein classes, we analyzed the derivative of Euclidean distances (Ex) between time points for all detected viral proteins (Fig. 7d). In parallel, we monitored their Tm values across time points. Not all viral proteins were detected at all time points, which was sometimes expected based on the temporal cascade of viral gene expression. Uneven detection may also derive from low protein abundance or ionization efficiency of tryptic peptides. Some viral proteins had relatively stable Ex and Tm values (e.g., immediate-early protein pUL123 and tegument protein pUS22), while other viral proteins displayed pronounced temporal changes, such as the kinase pUL97. pUL97 phosphorylates multiple host factors[61,62], including lamin A/C and the retinoblastoma protein, and viral proteins, including pUL44. The varied Ex values between time points reflect its multi-faceted roles and associations with substrates at distinct infection stages.

**Coaggregation data suggest virus–virus protein associations.** To search for possible virus–virus protein associations, we next extracted the melting curves of viral proteins and investigated their Ex values at each time point (Fig. 8a, Supplementary Fig. 12 and Supplementary Data 6). As expected, we identified a high Ex value for the immediate-early proteins IE1 and IE2 and similar aggregation profiles (Fig. 8b). These proteins are known to interact with pUL84[63] and complex with host proteins to inhibit immune responses. Similarly, high Ex values were observed for the capsid proteins MCP and SCP and the capsid-associated protein RIR1[17,64], and for known interacting tegument proteins pUL47 and pUL48[65]. These observations support the ability of this analysis to uncover viral protein associations. pUL47 and pUL48 are within a cluster (#1) of highly correlated viral proteins at 96 hpi. As these proteins localize at the virus assembly complex, our result may reflect the recruitment of other viral proteins to this compartment late in infection. Another cluster (#2) of highly correlated proteins was formed by pUS22 family members, including pUS23, pUS24, pTRS1, and pIRS1[66,67]. The correlated melting curves likely represent virion envelopment, and may point to unrecognized interactions within the virion.

**Essential viral pUL52 interacts with IFN-inducible proteins.** In addition to virus–virus protein associations, our dataset can help

uncover possible virus–host protein interactions. We readily observed the known pUL38 interaction with the TSC1/2 complex (Fig. 8c) that maintains mTOR activity and promotes viral translation[10]. The melting curves of these interacting proteins became closer late in infection (Fig. 8c). As our analysis captured known interactions, we next explored potential viral–host interactions not previously reported. Since the core-attachment-based method (COACH)[68] can help identify unknown PPIs[30], we used this algorithm to analyze our dataset (Supplementary Data 7). One of the complexes predicted at most time points contained pUL52, IFIT1, IFIT2, DDX58, and GOPC. The melting curves of the interferon-induced proteins with tetratricopeptide repeats (IFITs) IFIT1 and IFIT2 precisely overlapped in uninfected cells and at 24 hpi (Fig. 8d). At 48 hpi, the pUL52 melting curve was similar, but did not fully overlap with IFIT1 and IFIT2. However, at 72 and 96 hpi, these three melting curves became fully overlapped, suggesting their interaction late in infection. To confirm this association, we performed reciprocal IPs at 48 and 96 hpi. Endogenous IFIT1 co-isolated IFIT2 at both 48 and 96 hpi, while the level of associated pUL52 was less pronounced at 48 compared to 96 hpi (Fig. 8e). Although pUL52 is an essential HCMV protein, regulating the encapsidation of HCMV DNA[69], the specific mechanisms underlying its functions are not fully understood. As IFIT proteins function in immune response, it remains to be determined whether their association with pUL52 represents a viral immune-inhibitory strategy or a host antiviral response.

## Discussion

The spatial and temporal regulation of protein complexes and protein–protein interactions is fundamental to every step of an infection process, underlying both host defense and virus replication mechanisms. As a master manipulator of host cells, the ancient and large herpesvirus HCMV induces numerous alterations to protein functions, accomplished primarily via the temporal modulation of protein interactions. Identifying these functional interactions and capturing their dynamics is critical for understanding viral infection and for discovering therapeutic targets. Previous studies have benefited from the use of IP-MS for studying virus–host protein interactions[20]. These studies used isolations of viral or cellular proteins to identify direct and indirect interactions during infections, yielding important biological insights[10,70–73]. Other methods studying localization-specific PPIs have included proximity tagging[74,75], but their application to viral infections has been limited. Furthermore, one limitation of all these methods is their focus on the interactions of one protein of interest at a time and reduced capacity for high-throughput studies. Although recent efforts have pushed the limits of IP-MS to generate PPI

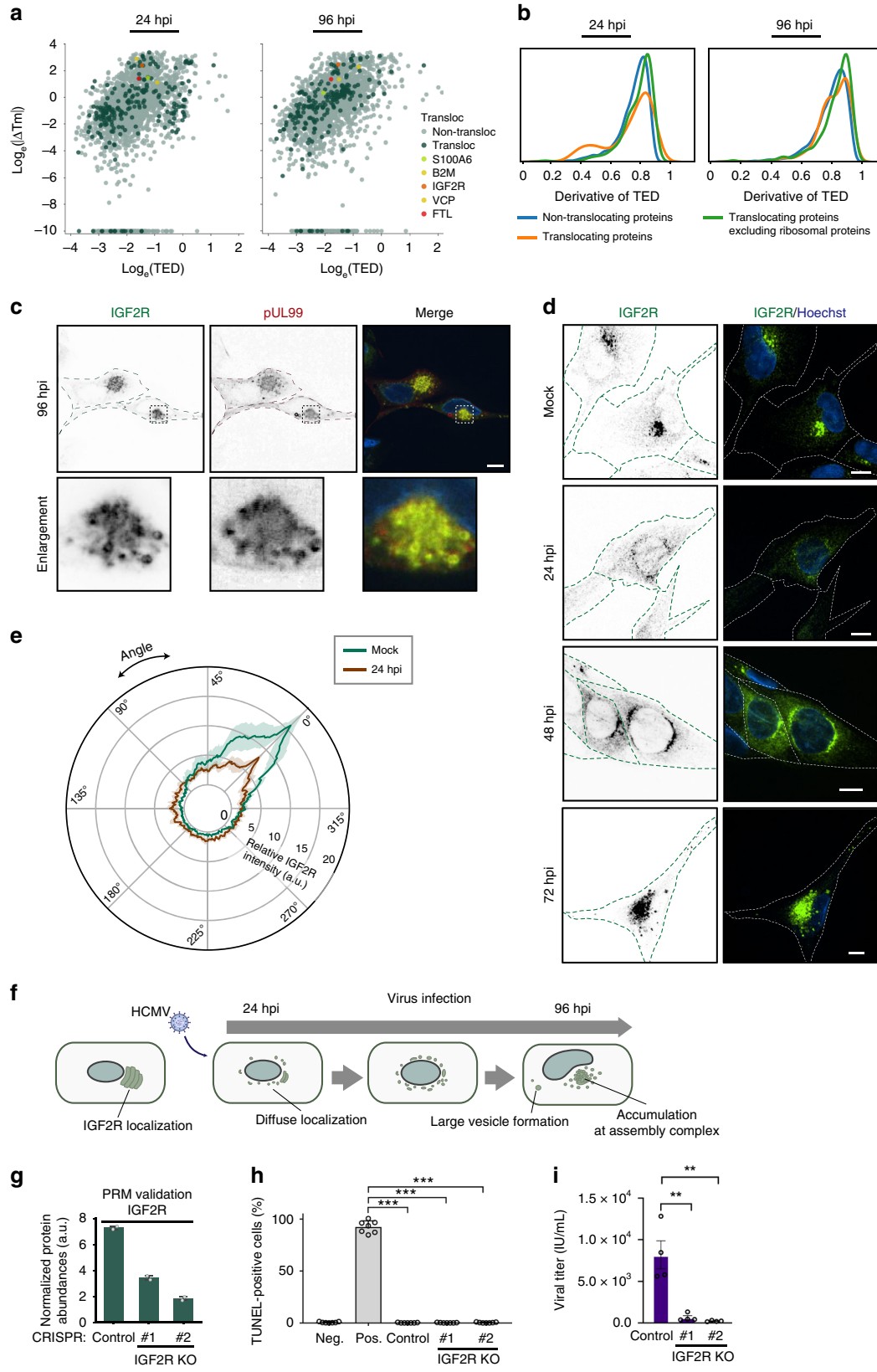

repositories[76,77], the large scale of these studies has so far allowed these to be performed in only a few cell types and biological conditions. Additionally, IP-MS and proximity labeling do not necessarily capture the dynamics of protein complexes, except when viewed from the perspective of the one protein of interest. A systems-wide view of protein complexes during an infection is still missing.

Here, we addressed this gap in knowledge using MS-based TPP and TPCA, in conjunction with microscopy and molecular virology. We monitored temporal alterations in protein

**Fig. 6 IGF2R is redistributed during infection and is needed for effective viral replication. a** Scatterplots of Tm values for all proteins in mock and infected conditions. Dark dots represent translocated proteins predicted by Jean Beltran et al.[12], and the dots in circles represent translocated proteins with substantial changes in Tm. **b** Distribution of the derivative of TED for non-translocating proteins, all translocating proteins, and translocating proteins excluding ribosomal proteins at 24 and 96 hpi. Translocating proteins are predicted from Jean Beltran et al. p-values calculated by a Mann–Whitney U-test on the distribution of TED values between non-translocating proteins and translocating proteins excluding ribosomal proteins are 3.76e-22 and 3.20e-21 for 24 and 96 hpi, respectively. **c** Immunofluorescence images of IGF2R and pUL99 at 96 hpi indicating co-localization at the viral assembly complex. Scale bar, 10 μm. **d** Immunofluorescence images of IGF2R at mock, 24, 48, and 72 hpi. Scale bar, 10 μm. **e** Polar plot of IGF2R fluorescence at mock and 24 hpi. Data are average fluorescence intensity ± SD (mock, n = 9; 24 hpi, n = 8). **f** Schematic of IGF2R dynamic redistribution throughout HCMV infection. IGF2R is localized next to the nucleus in uninfected cells. Upon infection, IGF2R puncta are dissipated around the nucleus, and large vesicles begin to form as infection progresses, resulting in IGF2R accumulation at the assembly complex late in infection. **g** PRM confirmation of IGF2R knockouts in CRISPR control and IGF2R-knockout MRC5 cells. Data represent average normalized protein abundance ± SD (n = 2). **h** TUNEL assay for IGF2R CRISPR knockout cells and CRISPR control cells, n = 6, ***p-value < 0.001. **i** Viral titers ± SEM from IGF2R CRISPR knockout cells and CRISPR control cells, a two-sided Student's t-test was performed, n = 4 biological replicates, **p-value < 0.01.

---

coaggregation profiles throughout HCMV infection, pointing to possible associations and dissociations between proteins (Fig. 9). Our results provided information regarding infection-induced changes in (1) known functional protein complexes critical for regulating cellular processes, and (2) interactions between individual proteins, including host–host, virus–host, and virus–virus PPIs.

Upon entry into cells, HCMV alters the levels of proteins at the plasma membrane and modulates cellular trafficking pathways[54]. Integrins are known as co-receptors for herpesvirus entry through association with the viral gB protein and the pentameric complex[48,78], facilitating HCMV internalization[48]. Indeed, we observed reduced ITGB1 associations with collagen and fibronectin, pointing to loss of plasma membrane interactions. Additionally, we uncovered another putative component of the ITGB1 internalization mechanism during infection. In conjunction with weakened ECM interactions, ITGB1 displayed enhanced association with the tetraspanin protein CD63. This agrees with a report of other tetraspanins' and integrins' roles during HCMV entry[79]. Furthermore, outside the context of infection, tetraspanins were observed to colocalize with ITGB1, suggesting involvement in ITGB1 internalization[80,81]. Our coaggregation and microscopy results highlighted increased ITGB1 association with CD63 starting at 24 hpi and becoming more pronounced at 48 hpi, concomitant with its internalization and degradation. Indeed, CD63 knockdown diminished ITGB1 internalization. This leads us to propose that ITGB1 is endocytosed with CD63 and subsequently degraded. PTMs could contribute to alterations in Tm values, subcellular localization, and protein abundances. For example, CD63 glycosylation was linked to localization changes[82]. Additionally, ubiquitinations and acetylations have been detected in ITGB1 during HCMV infection[83], with ubiquitination known to facilitate its degradation[84]. Given that some of these modifications were detected on the same residues, these PTMs may provide an on/off switch for ITGB1 degradation. Other proteins may contribute to this process, as we found increased ITGB1 association with CD47, a protein that interacts with the tetraspanin CD9 in uninfected cells[85]. CD63 localizes to late endosomes and lysosomes to mediate trafficking of extracellular vesicles (EVs) in uninfected cells, and CD63 is found in EVs released during HCMV infection[86]. Since we observed reduced virus titers upon CD63 knockdown, CD63 may play temporally distinct functions, supporting trafficking during virus assembly and egress late in infection.

By integrating our knowledge of protein localization during HCMV infection[12] and the translocatome dataset[55], we observed that many proteins predicted to translocate do not display altered aggregation profiles. It is possible that some translocation events occur via endosomal trafficking of complexes, rather than individual protein movements. However, in line with the understanding that protein localization can affect Tm[34,35], a subset of

known and predicted translocating proteins displayed significant Tm shifts during infection. Among these was the multifunctional receptor IGF2R, a protein involved in intracellular retrograde transport[87]. IGF2R was reported to support the replication of the gamma herpesvirus HHV-8 and the retrovirus HIV-1[88,89], and our results also establish it as a pro-viral factor during HCMV infection. While the disruption and reduction of IGF2R signal around the nucleus early in infection is consistent with the Golgi reorganization by HCMV[12], the IGF2R localization to the assembly complex late in infection and the inhibitory effect of its knockout on virus production suggest that HCMV hijacks IGF2R to coordinate cell signaling. IGF2R also plays a role in maturation of the alphaherpesvirus HSV-1 by binding to the glycoprotein gD during virion egress[90]. It is possible that IGF2R trafficking to the assembly complex during HCMV infection positions it to facilitate the maturation of HCMV glycoproteins.

Analysis of Ex and Tm values highlighted that viral proteins also display changing properties during infection, which can derive from altered localization, interactions, or PTMs. The distance matrix analysis captured known virus–virus protein interactions and provided a platform for predicting additional interactions between viral proteins. We observed a likely association between pUL71, a protein needed for virion secondary envelopment at the assembly complex[91], and the glycoprotein gH. pUL71 is conserved across herpesviruses, and its homolog in HSV-1 is involved in virion secondary envelopment and recruitment of the HSV-1 glycoprotein gE to the assembling virion[92]. Thus, our result points to a possible recruitment of gH to the maturing virion by pUL71. The analysis of virus–virus PPIs can also help illuminate the functions of unconventional HCMV proteins, such as ORFL147C. We identified a high Ex score for a putative interaction late in infection between ORFL147C and TRM3, a component of the tripartite terminase complex. This uncharacterized open reading frame (ORF) may function in viral genome packaging into the capsid. Similarly, our data predicted virus–host associations, including the interaction of the essential viral protein pUL52 with host immune factors.

Altogether, our study using thermal proximity profiling expands our understanding of protein complex dynamics during HCMV infection. Our results provide insights into functions of virus and host proteins, as well as a resource of temporal protein interactions, which we hope will inspire future PPI investigations during different types of viral infections. Looking ahead, there are aspects and limitations to be considered when using this approach for characterizing protein complexes, and improvements would be beneficial. First, careful bioinformatic assessment of the data is required to ensure robust quantitation and avoid false positives, as this approach requires integration of multiplexed datasets. Second, the simultaneous localization of a protein to multiple complexes may confound the interpretation of

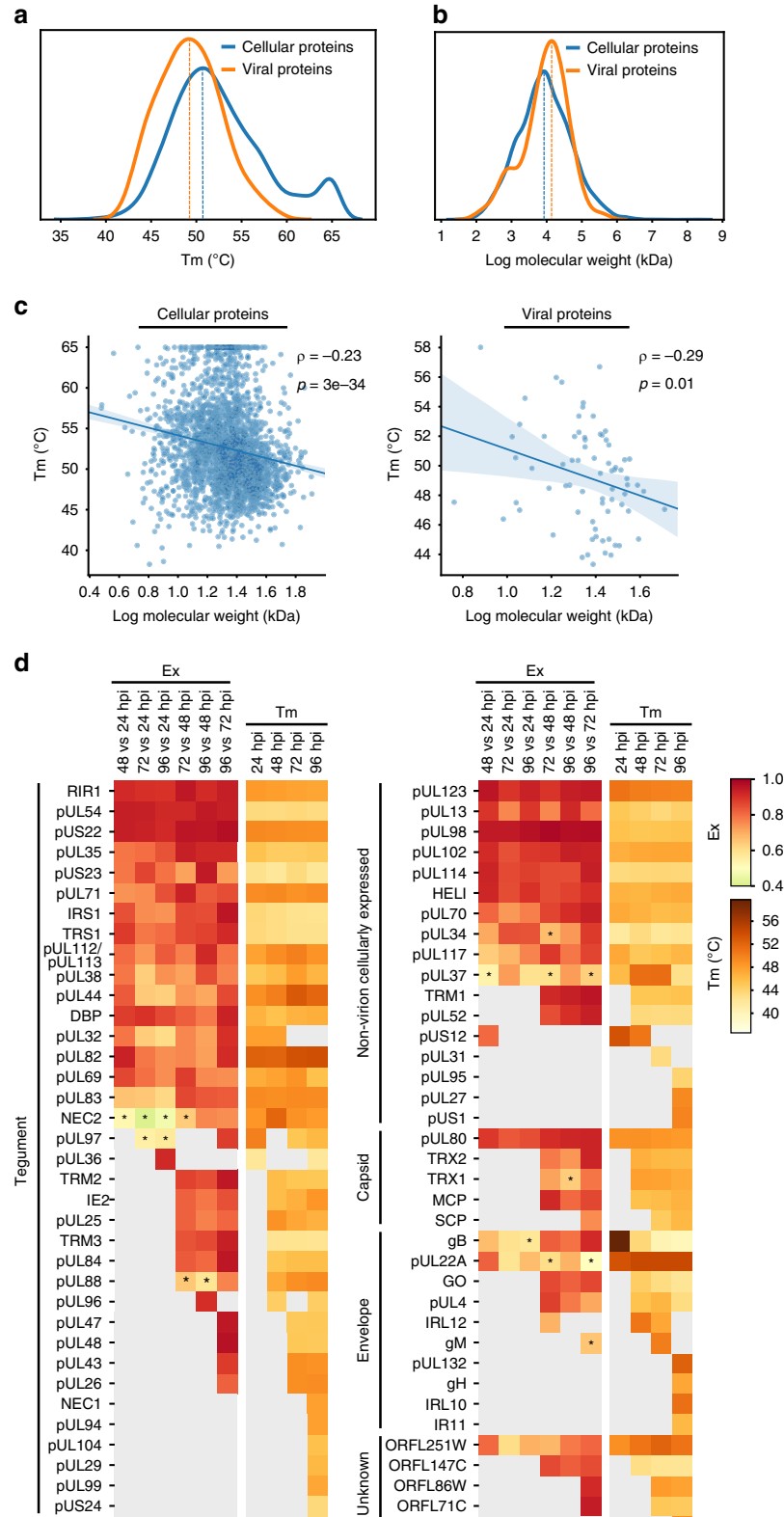

**Fig. 7 Viral proteins from different virion compartments display distinct temporal protein properties during infection. a** Distribution curves of Tm values for cellular proteins and viral proteins. **b** Distribution curves of molecular weights for cellular and viral proteins. **c** Linear correlation between log molecular weight and average Tm of cellular proteins (left) and viral proteins (right). Spearman's $\rho$ and $p$-values are shown. **d** Heat map of Ex and Tm values for all quantified viral proteins clustered by their compartment within the virion. Colors represent Ex or Tm values of the corresponding protein (see scale). Missing values in gray indicate a lack of quantification or inability to calculate Tm due to lack of a sigmoidal curve. Asterisks indicate $p$-values < 0.05 based on the distribution of Ex of indicated time pairs.

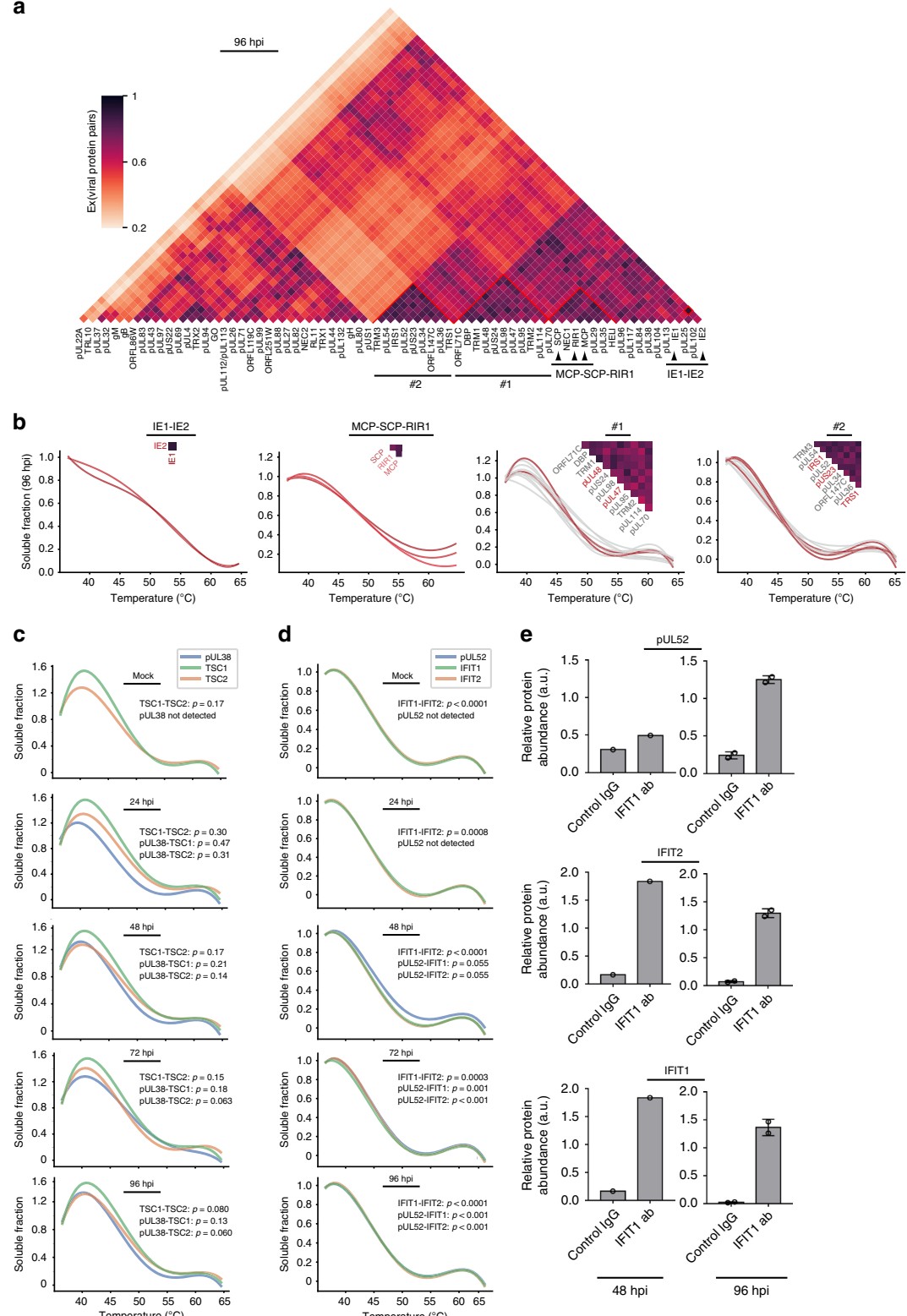

**Fig. 8 Thermal profiling points to virus–virus and virus–host protein interactions, including the association of the essential HCMV pUL52 with immunomodulatory host proteins. a** Distance matrix of Ex for pairs of quantified viral proteins. Color indicates Ex value for the corresponding protein pair. Clusters #1 and 2 highlight potential viral complexes. **b** Coaggregation curves of IE1 and IE2 (left), MCP, SCP, and RIR1 (left middle), proteins in cluster #1 (right middle) and proteins in cluster #2 (right) at 96 hpi. Known and literature-suggested complex components are indicated in red. **c**, **d** Coaggregation curves of **c** pUL38- TSC1-TSC2 and **d** pUL52-IFIT1- IFIT2 at each infection time point. pUL52 was not detected at mock and 24 hpi. Statistical analysis of curve similarities is shown with *p*-values (corrected with Benjamini-Hochberg procedure generated using random protein pair sampling). **e** IP-PRM analysis validating the complex composed of pUL52, IFIT1, and IFIT2. An anti-IFIT1 IP or a control IgG IP was conducted at 48 and 96 hpi and abundances of pUL52, IFIT1, and IFIT2 were quantified by PRM. Data represent average normalized abundance ± SD, *n* = 1 for 48 hpi, and *n* = 2 for 96 hpi.

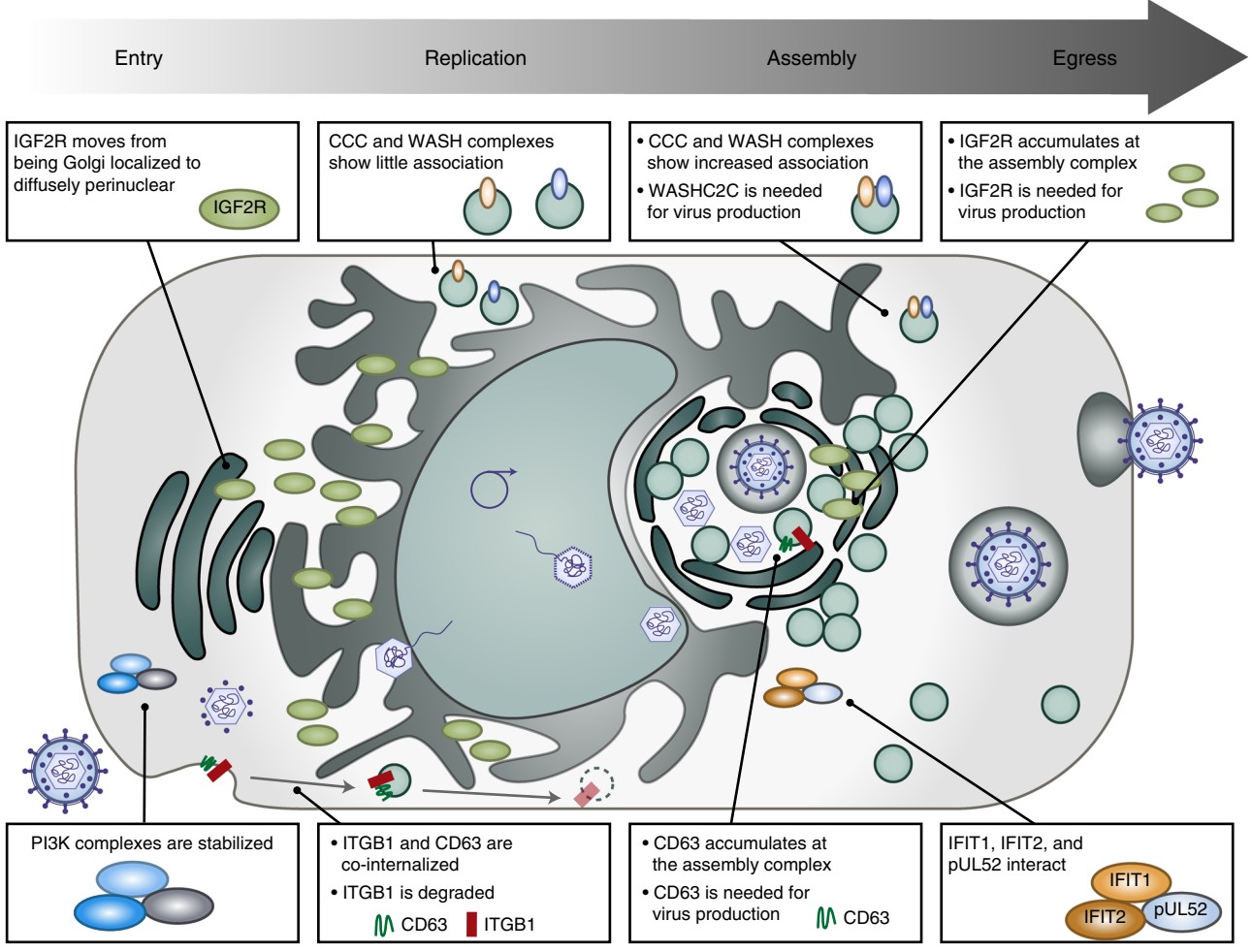

**Fig. 9 Thermal shift assays uncover temporal alterations in host–host and virus–host protein associations during HCMV infection.** At early stages of HCMV infection, the PI3K pathway is stabilized as the PI3K signaling axis is activated. Also early in infection, ITGB1, a receptor for HCMV, is internalized with the pro-viral tetraspanin CD63. As infection progresses, CD63 localizes to the viral assembly complex. Our study also uncovers a pro-viral role for the translocating protein IGF2R, whose localization changes from diffuse around the nucleus at early stages of HCMV infection to an accumulation at the viral assembly complex during virion maturation. Additionally, the endosomal transport complexes WASH and CCC display increased association late in infection, suggesting a pro-viral role via regulation of intracellular trafficking during viral assembly and egress. Finally, our study uncovers an association between the essential viral protein pUL52 and the host immune response proteins IFIT1 and IFIT2 at late stages of HCMV replication.

coaggregation curves, as the Tm values reflect the average of different protein states. Therefore, even if a protein is a component of a complex, its melting curve may not perfectly match those of the rest of the complex members. Nevertheless, our results showed that some multi-complex associations can be distinguished, e.g., pUL38 interaction with TSC1/2 and NuRD complexes[93]. Finally, extracting protein complex information largely relies on existing repositories. Despite these challenges, our results demonstrate that, in conjunction with functional assays, TPP approaches effectively expand our understanding of the dynamics of protein complexes and PPIs in the context of biological perturbations, including pathogen infections.

## Methods

**Cell culture and viral infection**. MRC5 (ATCC) and HEK293T (ATCC) cells were cultured under a humidified atmosphere of 5% $CO_2$ at 37 °C in culture medium Dulbecco's modified Eagle's medium (Life Technologies) supplemented with 10% (v/v) FBS (Invitrogen) and 1% (v/v) penicillin–streptomycin solution (GIBCO). Cells were tested for mycoplasma contamination by ATCC by agar culture, PCR-assay, and Hoechst DNA staining. MRC5 cells were used within ten passages. No additional cell line authentication was performed. HCMV strain AD169 was used as the wild-type strain for this study. Fibroblasts were infected in half volume media at a multiplicity of infection (MOI) 3 at 37 °C for 2 h. The media was then replaced with full volume fresh culture medium, and infected cells were cultured for the indicated length of time. Uninfected (mock) cells were also cultured in half volume media for 2 h, after which the media was changed to full volume fresh culture medium.

**Guide RNA (gRNA) and siRNA**. The following guide RNAs and siRNAs were used in this study. For CRISPR-mediated knockout of IGF2R, the guide RNAs used were CRISPR_IGF2R_1 5′-CTTACCTCTCTCCGCTCCGA-3′ (HGLibA_30096) and CRISPR_IGF2R_2 5′-GGCTTGTCCTGAGTTACGTG-3′ (HGLibA_30097). The control guide RNA used was NegativeControl (Origene pCas-Scramble) 5′-GC ACTACCAGAGCTAACTCA-3′. For siRNA-mediated knockdown of WASHC2C, the siRNAs used were SASI_Hs01_00233696 (Millipore Sigma) (5′-GAACCAAGC CUAGAACCAATT-3′) and SASI_Hs01_00233697 (5′-CAAACAAGAGCCGUGU CAATT-3′). For siRNA-mediated knockdown of ACAT1, the siRNAs used were 5′-GCGAAGAGGCUCAAUGUUA-3′ (Dharmacon J-009408-05), 5′-CUACAAGG CAGGCAGUAUU-3′ (Dharmacon J-009408-07), and 5′-GGAAGUGGUCAUAG UAAGU-3′ (Dharmacon J-009408-08), For siRNA-mediated knockdown of CD63, the siRNAs used were siCD63 #1 5′-GGAUGCAGGCAGAUUUUAATT-3′ and siCD63 #2 5′-GGAUUAAUUUCAACGAGAATT-3′. For siRNA-mediated knockdown controls, the siRNAs used were Universal negative control SIC005 and siGFP (5′-GGUGUGCUGUUUGGAGGUCUU-3′)

**Sample preparation for liquid chromatography–mass spectrometry (LC-MS/MS) analysis for TPCA analysis**. MRC5 cells were harvested from one confluent 15 cm dish ($6 \times 10^6$ cells) per condition. Cells were rinsed with 5 mL phosphate-

buffered saline (PBS) and detached with 2 mL trypsin (Life Technologies) followed by neutralization with culture medium. Cells were centrifuged at $250 \times g$ for 5 min, resuspended in 0.5 mL PBS, and aliquoted (50 μL/tube) into 10 PCR tubes. Each tube was heated at a different temperature for 3 min with a T100 thermal cycler (Bio-Rad). The temperatures used were the following: 36.9, 40.2, 43.9, 46.6, 48.6, 52.7, 55.3, 58.5, 61.2, and 64 °C. After heating, cells were incubated at 4 °C for 3 min. Next, cells were lysed in 100 μL kinase buffer (75 mM HEPES pH 7.5, 15 mM MgCl$_2$, protease and phosphatase inhibitor cocktail (Fisher Scientific, #78446), and 3 mM Tris(2-carboxyethyl)phosphine (TCEP)). The lysates were then frozen and thawed twice and mechanically lysed by (1) a 21-gauge needle with ten strokes and (2) a 26-gauge needle with six strokes. The lysates were frozen and thawed again, followed by centrifugation at $20,000 \times g$ for 20 min at 4 °C. The supernatants were reduced and alkylated with 25 mM TCEP and 25 mM chloroacetamide for 30 min at 55 °C and quenched with 25 mM cysteine. Proteins were precipitated with methanol and chloroform extraction and resuspended in 50 mM EPPS-KOH (pH 8.3). Protein concentrations were determined by BCA assay. The same volume of proteins from each heated sample was used for subsequent steps and was adjusted so that the sum of protein abundances of each infection time point was 1 mg. The protein samples were digested with trypsin (Thermo Scientific, #90059, 1:50 weight/weight) overnight at 37 °C. The digested peptides were adjusted to 100 mM EPPS (pH 8.3) and 20% acetonitrile, and labeled using a 10-plex TMT kit (Thermo Scientific, # 90111) for 1 h followed by quenching with 0.33% hydroxylamine. Two microliter peptides from each sample were acidified to 1% trifluoroacetic acid (TFA) and desalted with C18 StageTips as per manual (Fisher Scientific, #14–386–2). The resulting peptides were dried using a speedvac and resuspended in 5 μL of 1% formic acid/1% acetonitrile. Two microliter test mix of peptides was analyzed by LC-MS/MS analysis. To determine the appropriate mixing amounts, the median inversed scaling factor was used to fit the log-logistic function (optimize.curve_fit from Scipy v1.2) as follows: $LL(t) = c + \frac{1-c}{1+e^{b \cdot \log t - \log a}}$, where $t \in \mathbb{T} = \{36.9\,°C, 40.2\,°C, 43.9\,°C, 46.6\,°C, 48.6\,°C, 52.7\,°C, 55.3\,°C, 58.5\,°C, 61.2\,°C, 64\,°C\}$, and $a$, $b$, and $c$ are estimated parameters. Based on the fitted curve and the average protein abundances of all time points from each temperature calculated from the BCA assay, peptide concentrations were estimated, and the average volumes of samples to be mixed were determined so that the total peptide abundance in the mixed TMT labeled sample was 100 μg per sample. TMT samples were mixed with the average volume multiplied by each scaling factor. Mixed samples were adjusted to 1% TFA, incubated on ice for 15 min, and then centrifuged at $4000 \times g$ for 10 min at 4 °C. The peptides were eluted into 10 fractions using C18 StageTips. The elutions were dried and dissolved in a mixture of 5 μL of 1% formic acid and 2% acetonitrile, followed by LC-MS/MS analysis with 2 μL injection.

**LC-MS/MS analysis for TMT samples**. Peptides were applied to a nanoflow LC system (Dionex Ultimate 3000 nRSLC) equipped with an EASY-Spray ion source (C18, 50 mm × 75 mm, particle size of 2 mm). For the test mix, the peptides were fractionated with a linear gradient of solvent A (0.1% formic acid in water) and solvent B (97% acetonitrile, 0.1% formic acid, 2.9% water) from 6 to 18% solvent B over 60 min and with a linear gradient of 18 to 29% solvent B over 30 min, at a flow rate of 250 nl/min. Eluted peptides were sprayed directly into a Q-Exactive HF instrument (Thermo Fisher Scientific). MS and MS/MS spectra were obtained automatically in a full MS/data dependent MS2 mode. The full scan range was set to 350–1800 $m/z$ at 120,000 resolution, with an automatic gain control (AGC) target of 3e6, 30 ms maximum injection time (MIT), and spectra recorded in profile. Higher energy collision-induced dissociation (HCD) was performed on the top 20 most intense precursor ions with 25 s dynamic exclusion duration. The MS2 isolation windows were set to 1.2 $m/z$, with AGC target of 1e5, the MIT of 72 ms, a fixed first mass of 100 $m/z$, scan range 200–2000 $m/z$, resolution of 45,000, and spectra recorded in centroid. The analysis of fractionated TMT samples was performed identically as in the test mix analysis except for the MS2 isolation windows set to 0.8 $m/z$.

**LC-MS/MS analysis for targeted MS using PRM**. For targeted MS analysis, peptides were applied to the same LC-MS system that was used for the above TMT runs. Peptides were separated over a 60 min gradient (4 to 14% solvent B over 40 min, 14 to 25% over 20 min at a flow rate of 250 nl/min). One full duty cycle of the instrument consisted of one MS-SIM scan followed by 1 to 30 PRM scans. For full MS-SIM mode, the instrument was set to 350–1800 $m/z$ full scan range with a 15,000 resolution, 15 ms MIT, and 3e6 AGC target. For PRM scans, the instrument was set to 30,000 resolution, 80 ms MIT, 5e5 AGC, 0.8 $m/z$ isolation window, NCE of 28, and 150 $m/z$ fixed first mass. RAW files containing MS/MS spectra collected in PRM mode were imported into Skyline to extract product ion chromatograms (XICs) and calculate peak areas. A human and herpesvirus peptide spectral library was generated using Skyline (v4.0) from DDA MS analyses of Murray et al.[83] and Jean Beltran et al.[94]. The list of peptides was filtered based on ion intensity and peak area reproducibility across replicates. The top two to five most intense co-eluting product XICs were used for peak area quantification. Peak areas of IGF2R, WASHC2C, and CD63 peptides were normalized to peptides used as loading controls from HIST1H2BA (QVHPDTGISSK and LLLPGELAK), HIST1H2AA (AGLQFPVGR), TUBB1 (FPGQLNADLR) and GAPDH (LVINGNPITIFQER). For the WASHC2C IPs, peak areas of WASH complex proteins were normalized to those of the WASHC2C peptides.

**Peptide identification and quantification for TMT samples**. All MS/MS spectra were compared with protein sequences in the human and herpesvirus subset of UniProt-SwissProt database downloaded in April 2016 combined with common contaminants, using SEQUEST algorithm in Proteome Discoverer v2.2 (Thermo Fisher Scientific). The spectral recalibration node was utilized to perform off-line recalibration of mass accuracy. A fully tryptic constraint was placed on the database search, the allowed number of missed cleavages was set at 2, and static carbamidomethylation of cysteine and static TMT modification of N-terminus or lysine was set. Acetylation of NH2-terminal methionine, acetylation of NH2-terminus with methionine loss, oxidation of methionine, and deamination of asparagine were selected as dynamic modifications. Precursor mass tolerance was 5 ppm, and fragment mass tolerance was 0.02 Da. Matched spectra were evaluated by Percolator for false discovery rate (FDR) calculation (FDR of 1% was used) based on reverse sequence database searches. The integration tolerance and integration method of reporter ions quantifier node was set as 10 ppm and most confident centroid, respectively. In the consensus workflow, co-isolation threshold was 30 and average reporter $S/N$ was 8 for reporter quantification. Peptides containing deamidated asparagine or glutamine, phosphorylated serine, threonine, or tyrosine, and oxidized methionine were excluded from protein quantification.

**Processing of TMT data for normalization of soluble fraction**. To impute missing values of peptide abundances within TMT channels (temperature), we used the interval division based on points on both sides of the missing values with Gaussian noise. If the missing value is at 64 °C, the values were fitted with exponential decay (decay constant set as ln1.5) with Gaussian noise. Next, the abundance values of shared peptides from different proteins were distributed according to the number of peptide-spectrum matches (PSMs) used for quantification. For example, shared peptides were assigned to the proteins as in sp × $\frac{q}{Q}$, where sp is the TMT abundance of the shared peptide, $q$ is the sum of the spectral counts used for quantification at 37 °C of the assigned protein, and $Q$ is the total number of spectral counts for the peptide-sharing proteins. The abundance values of proteins were calculated as the sum of the peptide abundances that were quantified in all infection time points. For viral proteins, sequentially identified peptides were used for quantification. To obtain normalized solubility (df$_{i,r}$), protein abundances were divided by the abundance in the 37 °C sample, where $i$ and $r$ represent infection time point and replicate, respectively. df$_{i,r}$ is a $m \times 10$ matrix, where rows and columns represent proteins (total number of which is $m$) and temperature (36.9 °C, 40.2 °C, 43.9 °C, 46.6 °C, 48.6 °C, 52.7 °C, 55.3 °C, 58.5 °C, 61.2 °C, 64 °C), respectively. MoM$_{df_{i,r}}$ was defined as the 10-dimensional vector holding median values of normalized solubility at each temperature. MoMs$_i$ was defined as the median vector of MoM$_{df_{i,r}}$ between replicates. Adjusted solubility (cs$_{i,r}$) was defined as cs$_{i,r}$ = df$_{i,r}$ − 1 · (MoM$_{df_{i,r}}$ − f(MoMs$_i$))$^T$, where 1, $f(\cdot)$, and $T$ represent $m$ dimensional all-ones vector, 10-dimensional vector based on the inferred values of the fitting of log-logistic function LL($t$), and transpose operation, respectively. To normalize across replicates, we selected proteins that were identified in both replicates and performed weighted normalization based on the number of PSMs used for quantification as cs$_i$ = $\sum_{k=1}^{r}$ cs$_{i,k}$ ∘ (diag($U$)$^{-1}$ · $u_k$ · 1$^T$), where ∘, diag($\cdot$)$^{-1}$, $u_k$, $U$, and 1 represent the Hadamard product, the inverse of diagonal matrix, the spectral count detected in all infection time points in replicate $k$, the total number of spectral counts across replicates, and the 10-dimensional all-ones vector, respectively. To normalize between infection time points, we constructed the 10-dimensional median vector (MoM$_{cs}$) of MoM$_{cs_i}$, where MoM$_{cs_i}$ represents 10-dimensional median vector of cs$_i$. cs$_{norm_i}$ was defined as normalized adjusted solubility between both replicates at infection time $i$, where cs$_{norm_i}$ = cs$_i$ − 1 · $\left(\text{MoM}_{cs_i} - f(\text{MoM}_{cs})\right)^T$.

**Euclidean distance calculations**. Euclidean distance $d(p,q)$ is defined as $d(p,q)$ = $\sqrt{\sum_{t \in \mathbb{T}} (p_t - q_t)^2}$, where $p_t$ and $q_t$ represent normalized adjusted solubility of proteins $p$ and $q$ at heating temperature $t$. Therefore, we employed this calculation to assess the solubility differences between different proteins at one given time point. To assess the change in the solubility of a given protein across infection time points, we measured temporal Euclidean distances calculated as $TED(i) = \sqrt{\sum_{t \in \mathbb{T}} \left(p_{i,t} - p_{mock,t}\right)^2}$, where $t$ represents the heating temperature, and $p_i$ and $p_{mock}$ represent the normalized adjusted solubility of protein $p$ at an infection time point and uninfected state, respectively. Average Euclidean distance was defined as $\frac{\sum_{p,q \in c} d(p,q)}{\binom{|c|}{2}}$, where $c$, $|c|$, and $d(p,q)$ represent proteins identified as members of protein complexes in both CORUM and our TMT datasets, the number of proteins in $c$, and the distance between proteins $p$ and $q$, respectively. The average Euclidean distances were compared for CORUM complexes and random proteins (i.e., containing the same number of randomly selected proteins as those identified as putative CORUM complex members from our dataset). To better display the data, we also calculated the derivative of Euclidean distance as Ex = $\frac{1}{1+E_{avg}}$, where $E_{avg}$ is the average Euclidean distance of the protein pairs within the complex. Ex constrains the data range to (0, 1), and

it leads to a better fitting to a normal distribution compared to Euclidean distance.

**Scatter plot matrix generation.** Scatter plot matrices were created using matplotlib (v3.1) and seaborn (v0.9) Python packages. Proteins (TMT channels) or protein complexes with missing values in each sample were removed from the scatter plot. For the generation of the scatter plot of relative protein abundances (Fig. 1b and Supplementary Fig. 1b), the correlation coefficients between replicates were calculated as $\log_e$ transformed protein abundances, estimated from the sum of the peptide reporter ion intensities for peptides shared across time points for the indicated proteins. For the scatter plot of protein complexes (Fig. 1g), a derivative of average Euclidean distance (Ex) was used and Ex was calculated as indicated above. The correlations were calculated using Pearson's $r$. The height and width of ellipse behind the correlation were calculated as $\sqrt{1-r}$ and $\sqrt{1+r}$, respectively. The heat map was colored according to $(r-0.8)\times 5$.

**PCA analysis.** Normalized protein abundances in TMT channels per time point were standardized with StandardScaler of the scikit-learn (v0.21) Python package. Resulting normalized channels of each time point were then analyzed by the PCA function of scikit-learn. Percentage of variance explained by each of the selected components was calculated with explained_variance_ratio_ of scikit-learn. Plotting of the proteome abundance data in PCA space used abundances calculated as the sum of all peptide reporter ion intensities for each protein.

**Fuzzy c-means clustering and GO enrichment analysis.** To check complex alterations after infection, we used $\text{Ex}_{c,i}$ ($=\text{Ex}_{c,i}-\text{Ex}_{c,\text{mock}}$), where $\text{Ex}_{c,i}=\frac{1}{1+E_{c,i}}$ and $E_{c,i}$ is the average Euclidean distance of each protein within the complex ($c$) at specific infection time point ($i$; 24, 48, 72, and 96 hpi or mock). $z$-scores were calculated from the null-distribution of $\text{Ex}_{c,i}$ generated from random proteins of the same number as in complex ($c$) with 10,000 iterations. Clustering of the pathways were performed with the fuzzy c-means clustering based on time series of $z$-scores from SciKit-Fuzzy (v0.4) python package. Fuzzy c-means parameter m and error were set as 2 and 0.005, respectively. Cluster number was determined by final fuzzy partition coefficient. Membership scores were calculated as $\max_j(\mu_{jl})$ of the final $\mu_{jl}$ ($0\le j\le C$, $0\le l\le N$), where $N$ and $C$ represent the number of datasets (protein complexes) and clusters ($C$ set to 8). The final fuzzy partition coefficient $\mu_{jb}$, ranging from 0 to 1, is calculated via iterative optimization, and it represents the harmonic mean of the relative distance between complex $l$ and the center of cluster $j$ with a weighting fuzziness parameter $m'$ ($m'$ set to 2). $\mu_{jl}$ depicts the chances of protein complex $l$ being assigned to cluster $j$. Altogether, the membership scores informed of the most likely cluster a protein complex $l$ is assigned to, and the relative distance of this complex from the average (centroid) of the assigned cluster. GO terms were retrieved from the CORUM database and used for enrichment analysis. Enrichment analysis was carried out by Fisher's exact test. The enrichment score was defined as $-\log_{10}p-$value, which was calculated by scipy.stats.hypergeom of Scipy. Terms with enrichment scores >1.5 were selected. All the enriched GO terms, without a limit to the number of complexes assigned to a GO term, are displayed.

**Protein complex analysis and data representation.** Proteins quantified in our datasets were searched in the CORUM database, which contains curated human protein–protein complexes that have prior experimental validation from the literature. The information regarding putative complexes present in our dataset was used to determine whether the complex members have similar aggregation profiles in fibroblasts, suggesting their possible association. For this, average Euclidean distances were calculated between the protein aggregation curves of the members of predicted complexes. This calculation was repeated at each infection time point to determine whether these putative associations change during the progression of infection. For Fig. 1f, a Gaussian distribution was fitted to the distribution of the derivative of average Euclidean distance of putative protein complexes among randomly selected proteins, and the $z$-score describes the confidence in the different mean values of the distributions of the CORUM complexes and the randomly selected proteins. $z$-score describes the number of standard deviations of the CORUM complex proteins from the mean of the randomly selected proteins. High $z$-scores represent high Ex and low-average Euclidean distance, which indicates closer melting curves of the complex components. For network illustration of protein complexes of interest, the edge is defined as the interaction between protein nodes as documented by the CORUM database. Edge width and color represent the Ex value of interacting proteins (nodes) and $z$-score derived from the Ex values, respectively. The network plots were constructed with NetworkX (v2.3) python package and Cytoscape (v3.7).

**Immunoaffinity purification sample preparation.** MRC5 cells were mechanically lysed with a Polytron in lysis buffer (20 mM HEPES-KOH pH 7.4, 110 mM potassium acetate, 2 mM MgCl₂, 0.1% Tween-20, 1 μM ZnCl₂, and 1 μM CaCl₂) containing 0.2% Triton X-100 and protease and phosphatase inhibitor cocktail. Protein A/G magnetic beads (Pierce #88802) were washed with the lysis buffer and conjugated to the following antibodies for 1 h at 4 °C; anti-FLAG antibody

(AB_262044), anti-IFIT1 antibody (AB_2783869), or purified rabbit IgG (MP Biomedicals). Cell lysates were incubated with the conjugated beads for 1 h at 4 °C. Beads were washed with 0.5 mL lysis buffer three times, followed by one wash in cold MilliQ water. Proteins were eluted with 50 μL TES buffer (106 mM Tris HCl, 141 mM Tris Base, 2% sodium dodecyl sulfate (SDS), 0.5 mM EDTA) for 10 min at 70 °C, and concentrated 2.5 times by speedvac. The samples were reduced and alkylated with 25 mM TCEP and 25 mM chloroacetamide for 20 min at 70 °C. Samples were prepared for S-trap columns by adding 2.3 μL 12% phosphoric acid and 165 μL 100 mM TEAB (pH 7.1) in methanol. Samples were then applied to the S-trap column (Protifi, C02-micro) and washed with 150 μL 100 mM TEAB pH 7.1 in methanol five times. Protein samples were digested with trypsin solution (2 μg trypsin in 25 mM TEAB pH 8.0) on column for 1 h at 47 °C. The resulting peptides were eluted sequentially with 40 μL 25 mM TEAB pH 8.0, 0.2% formic acid, and 0.2% formic acid in 50% acetonitrile. The eluted peptides were dried and resuspended in 1% formic acid/2% acetonitrile and subjected to PRM analysis as described above.

**Enrichment analysis using |ΔTm|.** Tm value is designated as the temperature at the inflection point of the fitted log-logistic function LL($t$), which was determined as the parameter $a$ as described above. Absolute ΔTm values of indicated infection time points were sorted from high to low, and the values were evenly divided into seven bins. Then enrichment analysis based on information theory was performed[95]. Heat map colors of bins represent the enrichment scores and were plotted as $-\log_{10}p$-value based on Fisher's exact test. $z$-scores were calculated based on the null-distribution of mutual information of the indicated gene sets and were derived from 10,000 iterative random samplings.

**Immunofluorescence analysis.** For ITGB1 and CD63 immunostaining, MRC5 cells on coverslips were washed with 0.5 mL PBS twice and fixed in 4% PFA at room temperature for 15 min. Fixed cells were washed with 0.5 mL PBS three times, permeabilized with 0.2% Tween-20 in PBS for 10 min at room temperature, and blocked with blocking buffer (5% goat serum, 5% human serum, 22.5 mg/ml glycine, 3% bovine serum albumin (BSA), 0.2% Tween-20 in PBS) for 30 min at room temperature. Samples were incubated at room temperature for 2.5 h with primary antibodies to ITGB1 (12G10, Santa Cruz 1:150) or CD63 (ab118307, abcam 1:100) diluted in 3% BSA in 0.1% TBST, followed by incubation at room temperature for 45 min with secondary Alexa Fluor 555- or Alexa Fluor 488-conjugated goat antibodies to mouse or rabbit IgG (AB_141780 or AB_143165, Life Technologies, 1:1000, respectively) diluted in 3% BSA in 0.1% TBST. Samples were then stained with Hoechst 33258 (1:1000) for 15 min at room temperature. For IGF2R and pUL99 immunostaining, the same method was applied except that the samples were fixed in 100% methanol at −20 °C for 15 min and that primary antibodies to IGF2R (D3V8C, Cell Signaling, 1:150) and pUL99 (gift from Thomas Shenk, clone 10B4, 1:150) were used. Cells were mounted with ProLong Diamond Antifade Mountant (Thermo Fisher Scientific, P36965) on glass slides after staining and examined with an inverted fluorescence confocal microscope (Nikon Ti-E) equipped with a Yokogawa spinning disc (CSU-21), digital CMOS camera (Hamamatsu ORCA-Flash TuCam), and precision microscope stage (Piezo). Mean intensity of ITGB1 or CD63 in the region of interest (ROI) was measured by Fiji.

**Western blot analysis.** Samples were lysed in lysis buffer (50 mM Tris-HCl pH 8.0, 100 mM NaCl, 0.5 mM EDTA, 4% SDS) and subjected to iterative rounds of heating at 95 °C and cup horn sonication until lysis was complete. Protein concentration was determined by BCA assay, and equal amounts of proteins were loaded on a 10% Tris-glycine SDS-polyacrylamide gel that was run at 130 V. Proteins were transferred overnight at 4 °C at 30 V onto a polyvinylidene fluoride (PVDF) membrane. Membranes were cut and blocked for 1 h in blocking buffer (5% milk, 0.2% Tween-20 in PBS). Membranes were incubated with primary antibodies in block for 3 h at room temperature: ACAT1 (1:1000, PA5-82154, Thermo Fisher Scientific), IE1 (1:500, clone IB12, gift from Thomas Shenk), pUL26 (1:200, gift from Thomas Shenk), pUL99 (1:200, clone 10B4, gift from Thomas Shenk), and tubulin (1:5000, T6199 clone DM1A, Sigma). Membranes were incubated for 1 h with secondary antibodies in block: Alexa 680 and 800 anti-mouse or anti-rabbit IgG (H+L) (Invitrogen, 1:10,000). To determine knockdown efficiency, densitometry analysis was performed using the ImageStudioLite for Odyssey. Intensities of ACAT1 were normalized to those of tubulin.

**IGF2R intensity distribution analysis.** Images were analyzed in Fiji. We subtracted the background intensity with a threshold of 150. For each cell, we selected ROI, and removed the outside intensity. Center of the ROI was determined as the center of the nucleus, and the image was polar transformed using Fiji plug-in, Polar Transformer. An intensity histogram of the transformed image was generated by the Plot Profile function in Fiji. To construct the polar plot, the maximum intensity of each image was aligned to 0°, and the polar plot was drawn using matplotlib Python package.

**Generation of CRISPR knockout cells.** CRISPR-mediated knockouts of IGF2R or control knockouts were performed with the guide RNA sequences indicated in the gRNA and siRNA section. Guide RNA sequences were inserted to LentiCRISPRv2

(a gift from Feng Zhang, Addgene # 52961). Generated lentivirus was used to transduce MRC5 cells, which were selected by puromycin (2.5 μg/mL; Life Technologies).

**Viral titer analysis**. MRC5 cells were infected with HCMV at MOI 3, and supernatants were collected at 120 hpi. The supernatants were used to infect a reporter plate of wild-type MRC5 cells. After 24 h, the infected cells were fixed with methanol, blocked with blocking buffer (3% BSA, 0.1% Tween-20 in PBS) for 1 h, incubated with anti-IE1 antibody (a gift from Thomas Shenk, clone IB12, 1:100) for 1 h, and incubated with Alexa 488 anti-mouse goat IgG (1:1000; Life Technologies, A-11001) for 1 h. Samples were then stained with Hoechst 33258 (1:1000) for 15 min. Viral titers were calculated with the Operetta imaging system (Perkin Elmer). The number of IE1-positive cells was used to calculate infectious units/mL (IU/mL).

**Protein complex detection using COACH algorithm**. The PPI network was constructed from the derivative of Euclidean distance ($Ex = 1/(1 + E)$) of each protein pair, and the network edge was set as Ex with a threshold of 0.9. The COACH algorithm[68] was applied to this network, and the parameters $d$ and $t$ were set as 0.7 and 0.225, respectively. To investigate putative protein complexes at different infection times, we selected core complexes as a complex subset based on COACH with the Jaccard index >0.34, and pUL52 containing complexes were searched within the subset.

**siRNA-mediated knockdowns**. Knockdowns were carried out by transfecting MRC5 cells with siRNAs (see gRNA and siRNA section for sequences) for ACAT1, WASHC2C, and CD63. For titer analysis, MRC5 cells were cultured in a 12-well plate and were transfected with 30 pmol of the targeting siRNAs or control siRNA using Lipofectamine RNAimax (Thermo Fisher Scientific). Cells treated with siRNAs were infected with HCMV 2 days post siRNA transfection and re-treated with the siRNAs on the same day of infection to ensure that the genes of interest stayed knocked down throughout the viral replication cycle. Supernatants were collected at 120 hpi and used for viral titer analysis as described above. To validate the efficiency of the WASHC2C and CD63 knockdowns, cells were collected at 48 h post transfection and subjected to PRM analysis as described above. To validate the efficiency of the ACAT1 knockdowns, cells were collected at 72 h post transfection and subjected to Western blotting. To assay the effect of WASHC2C or CD63 knockdowns on virus replication, cells were transfected with the siRNAs and infected 48 h later. When infection media was changed after the 2 h incubation, the cells were re-treated with siRNA to ensure knockdown was maintained throughout infection. To assay the effect of ACAT1 knockdowns on virus replication, transfected cells were infected with HCMV at MOI 3 at 72 h after transfection, and the supernatants at 120 hpi were collected and used for viral titer analysis, as described above.

**TUNEL assay**. To evaluate cell death induced by apoptosis after siRNA treatment or CRISPR knockout, an in situ cell death detection kit (Sigma Aldrich) based on TUNEL (terminal deoxynucleotidyl transferase dUTP nick end labeling) staining was used for quantification per manufacturer's guidelines. Briefly, cells in six replicates were cultured in a 96-well plate and analyzed 48 h after siRNA treatment. Cells were washed with PBS and fixed with 4% paraformaldehyde in PBS for 20 min at room temperature. Fixed cells were permeabilized with 50 μL of 0.1% Triton, 0.1% sodium citrate in PBS for 2 min at 4 °C, followed by two washes with PBS. Samples were then incubated with 25 μL TUNEL mix for 1 h at 37 °C in the dark. For positive controls, samples were treated with 10 μg/mL DNase I in 50 mM Tris-HCl (pH 7.5) with 1 mg/mL BSA for 10 min at room temperature to induce DNA strand breaks; omission of enzyme solution provided the negative control. Samples were washed with PBS for 5 min three times, incubated with 0.1 μg/mL DAPI for 10 min, and washed with PBS for 5 min three times. Images were taken by Operetta imaging system (Perkin Elmer) to quantify the percentage of TUNEL positive cells.

**Protein abundance normalization during infection**. Protein abundances during infection were based on the abundances in the soluble fraction at 37 °C from this study. To normalize protein abundances between infection time points and replicates, we used the median normalization method using NormalizerDE. Median abundances of replicates were used as normalized protein abundances at each time point. To calculate protein expression changes during infection, the ratio of mock and indicated infectious time of normalized abundances were determined in log space.

**Calculation of NES change during infection**. To calculate NES based on the distribution of Ex, we performed Gene Set Enrichment Analysis (GSEA)-based enrichment analysis. The difference in empirical cumulative distribution functions (ΔECDF) of TED in all quantified proteins and annotated proteins of focusing gene set were used to calculate enrichment score as $\int|ECDF|$. NES is determined as $\frac{s \int|ECDF|}{\left(\frac{1}{n}\right)\sum_{i=1}^{n}\int|ECDF_{random_i}|}$, where $s$ represents 1 or –1 as described below, and where $ECDF_{random}$ is calculated in all quantified proteins and randomly selected proteins,

which are the same number of proteins to the focusing gene set. n represents the number of random selection and we set $n = 1000$. If $\int ECDF > 0$, $s$ represents 1, otherwise $-1$. To determine the most modulated pathways by NES during infection, we used the values of (max (NES during infection time) – min (NES during infection time)) and sorted them from higher value. Distributions of Ex histograms of all annotated proteins or the indicated gene set were fitted by kernel density estimation using the function of gaussian_kde from statistical functions of Scipy.

**Assignment of translocating proteins**. Translocating proteins during HCMV infection were determined based on Jean Beltran et al.[12]. We defined a translocating protein as one that has different predicted subcellular localization between uninfected and any infected conditions. In the translocatome dataset[55], high-confidence translocating proteins were determined by over 0.6167 Translocation Evidence Score (TES) values.

**Molecular weight, GRAVY, CvP bias calculation**. Molecular weight and GRAVY were calculated by The Sequence Manipulation Suite. CvP bias were calculated as $\{D + E + K + R - (N + Q + S + T)\}/(protein\ length) \times 100$, where $D$, $E$, $K$, $R$, $N$, $Q$, $S$, and $T$ are the number of indicated amino acid in the protein. Distributions of histograms were fitted by kernel density estimation using the function of gaussian_kde from statistical functions of Scipy.

**Generating distance matrices of virus proteins**. Euclidean distance matrices of virus proteins were clustered by average (unweighted pair group method with arithmetic mean; UPGMA) method of Scipy. The heat map color represents the derivative of Euclidean distance $1/(1 + E_{virus})$, where $E_{virus}$ represents Euclidean distance of indicated virus protein pair.

**Statistics**. Statistical analyses were performed using a two-tailed student's $t$-test unless otherwise stated. Data are presented as mean ± SD or mean ± SEM, as indicated. The number of biological replicates is indicated with "$n = $".

**Reporting summary**. Further information on research design is available in the Nature Research Reporting Summary linked to this article.

## Data availability

Proteomics data were deposited to the ProteomeXchange Consortium via the PRIDE partner repository with the dataset identifier PXD014747. Source data underlying Figs. 3g and 5g are provided as a Source data file. Any additional data are available from the corresponding author upon reasonable request.

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

## Acknowledgements

This research was supported by funding from the NIH (GM114141) and a Mallinckrodt Scholar Award to I.M.C., a National Science Foundation Graduate Research Fellowship (NSF-GRFP DGE-1656466) to L.A.M.N., and the NIH NIGMS (T32GM007388).

## Author contributions

Y.H. and I.M.C. designed research. Y.H., X.S. and L.A.M.N. performed experiments. Y.H., X.S. L.A.M.N. and I.M.C. analyzed data. Y.H., X.S., L.A.M.N. and I.M.C. wrote, edited, and reviewed the manuscript.

## Competing interests

The authors declare no competing interests.
