## [Peer Review File · Nature Communications]

Reviewers' comments:

Reviewer #1 (Remarks to the Author):

Hashimoto and colleagues had done a remarkable work deploying cellular thermal shift assay and quantitative mass spectrometry with the recent TPCA (thermal proximity co-aggregation) methodology to study the dynamics of protein complex in primary human fibroblast during HCMV infection. In addition to the approaches in Tan et al (2018), I am glad they had extended the methodology to include multiple time points and inter-species comparison that were accompanied by new analysis workflow. Their work and data provided a system view of proteins and interactions that are post-translationally modulated at different stage of HCMV infection. Quality of their work, data and analysis is exemplified by validation of multiple modulated complexes identified with Co-IP and microscopy (co-localisation), and the linkage of the associated proteins to virus production through siRNA.

The authors have also made conceptual advances showing that similar co-aggregation profiles of their data can be used to identify novel association among small group of viral proteins. For a large set that consist of both viral and human proteins, they demonstrated that their data can be used to construct a network of proteins with high co-aggregation similarity that was used with the COACH algorithm to identify potentially novel viral-host protein complexes. In particular, they experimentally validated novel interactions between viral protein pUL52 with IFIT1 and IFIT2 proteins inferred from such analysis. The manuscript is also very well-written and organised. What I think is lacking is a final figure depicting the virus infection process but highlighting the key findings or observation from this work. Otherwise, below are my comments and suggestions that could be useful for improving the work further.

Major

1. The authors had highlighted that the T_m of a subset of proteins could not be computed as they display non-sigmoidal curves, thus had used a complementary measure, difference in area under the curve (AUC), to identify more differentially modulated proteins. Nevertheless, it is possible that the shape of a curve could change with minimum difference in AUC. Euclidean distance between the curves of the same protein at different time point could be used to detect such changes although the measure lacks information on the direction of change. I will appreciate if the authors could do a comparative analysis whether there is substantial difference between AUC or Euclidean distance for identifying differentially modulated proteins in HCMV infection.

2. On the observation that most protein known to or potentially translocate do not exhibit much difference in T_m values, is the observation similar based on AUC or euclidean distance? How much is

known about the portion of these proteins that translocate? If only a small portion translocate, the non-translocating subset will dominate the signal. In addition, for known translocating proteins with observable difference in T_m , are there enrichment in any organelle or cellular location?

Minor

3. Line 107: As it not clear that proteins in the same complex need to unfold similarly (it may be well possible a subunit could unfold first to induce precipitation) to have similar (in)solubility, perhaps the sentence could be edited to "This method is based on the principle that proteins in the same complex exhibit similar progressive insolubility when treated with an increasing temperature gradient"

4. Table S1C: Are the values in the table MS2 intensity of reporter ions? If so, the use of word "Abundances" in description (sheet 1) is potentially misleading.

5. Line 148: Please kindly clarify how the correlation coefficients (for reproducibility analysis) are computed. Are solubility ratios or raw reporter intensities used?

6. Supplementary Fig 1b. It is unclear what are the axis of individual scatterplot: are they log10 or log2 of reporter ion intensity or something else?

7. Line 151: Fig 1c depicts increasing number of viral protein being identified as infection progresses that may not necessary suggest increased abundance of individual viral proteins. For viral proteins identified across all time points (except mock, of course), maybe the changes in abundance of each protein can be expressed as fraction at 24hrs and the data distribution visualised as a box plot.

8. Fig 1f: Shouldn't the legend be

Z score:>2.0

Z score:1.5 to 2.0

Z score:1 to 1.5

9. Line 151-154: Please kindly clarify whether abundance ratio or raw reporter intensity is used?

10. Fig 3b. Maybe the subunits(node) of CCC complex and WASH complex can be uniquely coloured.

11. Fig 8d: Might be helpful to explicitly indicate on the diagram or caption that pUL52 is not detected in Mock and at 24hpi.

12. Line 771-773: It is unclear why for the GO enrichment analysis, the minimum number of complexes per GO term need to be set at 6? If the set is small, signal will not be significant more often than not. Is this for reason of multiple hypothesis correction?

13. Table 2A: The subunits of each protein complexes with curves should be listed.

Reviewer #2 (Remarks to the Author):

The manuscript entitled “Temporal dynamics of protein complex formation and dissociation during human cytomegalovirus infection” by Hashimoto et al utilizes a thermal shift assay paired with mass spectroscopy to define the cellular protein interactions that occur during a viral life cycle. The strategy is quite innovative and has the potential to reveal important protein interactions that are critical for virus infection, replication, and dissemination. The manuscript describes several protein interactions CCC-WASH complex and ITGB1-CD63 as well as other cellular factors that have been demonstrated to play a role in virus infection. The findings were also validated by siRNA data studies to demonstrate the role of the identified protein. Even though most of the protein identified in this study have been shown to play a role in CMV infection in previous studies, the strategy to identify these protein interactions and validation of the findings is quite novel. However, there are some technical issues that would enhance the manuscript for a reader not well versed in mass spectroscopy.

- 1) The analysis of protein complexes should be better described in the Results Section. A diagram of how protein-protein complexes were identified should be included.
- 2) The manuscript should better describe the relevance of Euclidian distances and how they can be derived from the mass spec data. In fact, the Average Euclidian distances in Figure 1e and CORUM complexes in Figure 1f do not show significance differences. Is this typical data?
- 3) All of the bar graphs should include a statistical analysis.
- 4) As represented, Figure 3b is very confusing. The Figure should be modified.
- 5) Figure 3c IP-PRM analysis is not very convincing. Can this be improved?

6) The viral titer validation assays are a little confusing because the inclusion of siRNA would knock-down the gene for several days, but a virus titer assay is typically 10-14 days. This should be clarified.

Reviewer #3 (Remarks to the Author):

Comments to authors:

In this work, Hashimoto et al. generate a temporally-resolved dataset of protein thermal stability and abundance during viral infection. This resource on its own is potentially valuable for the research community and might support hypotheses from other research groups. The authors go through some length to try to link some of the observations to known biology and potentially find hints of new molecular targets to fight infection. However, I think some of those experiments lack proper controls. Overall, the manuscript is well written and generally clear in its message, although some of the concepts/terms used might not be easily understood by a broad audience and should be more clear. I offer some comments that I think would greatly improve the manuscript.

Major comments:

1. As a general comment, the authors focus most of the manuscript on protein-protein interactions. While this is interesting, there are many reasons for changes in T_m of proteins that are unrelated to direct protein interactions (e.g., binding of small molecules (metabolites), binding to nucleic acids, changes in protein localization, PTMs). Some of the apparent reduced/increased distance between protein pairs might be just by chance, because one of the proteins is engaged in one of the above mentioned interactions. When I looked into some melting curves of protein pairs discussed in the manuscript, many times the pairs that got closer do not always completely overlap, as would be expected if they are co-aggregating. Also, some times the pair got closer by the thermal destabilization of one of the proteins. I have trouble rationalizing why this would happen, i.e., if a protein is more thermostable when alone, why would it become less thermostable when engaged in an interaction. The authors should be more careful in their overinterpretation of the data and as a minimum inform the reader that some of the changes might not be related to PPIs. Thermal proteome profiling has been shown to inform on other processes such as e.g. protein metabolite interactions and pathway activity (PMID: 30858367, PMID: 29980614, PMID: 29706546). The possibility of such interpretations should be taken into account and at the very least mentioned. I would suggest that the authors look more carefully into the function of the proteins discussed to ensure that neither of the other mechanisms is likely to cause the changes in thermal stability. It would also be helpful to have supplementary figures with the melting curves of the pairs of proteins described.

2. Related to the previous comment, although it would require major rearrangement of the manuscript, I think a more logical way to tell this story would be to look first at ΔT_m or ΔAUC

throughout infection and which proteins/processes are changes. Then, try to rationalize some of these changes, which in part would reflect the spectrum of PPIs changing, but also many other types of interactions/biophysical changes to the protein. Related to that, the original name of the combination of CETSA with quantitative mass spectrometry, “thermal proteome profiling, TPP”, should be mentioned, since this is what is used in the manuscript.

3. Throughout the text, the authors compare all their results to a “mock” infection. I could not find in the methods section what this means (from methods section: line 597 “The media was then replaced with fresh culture medium, and infected cells were cultured for the indicated length of time.”). What is the time for the mock infection? For me, a better and important control would be a 0h time point (i.e., cells would go through the 2h initial infection and then medium change). If this is too long and too many things already happen in that time period, at least the cells should go through the same medium exchange that the infected cells go through, but without the virus.

4. The results of the interaction between the WASH and CCC complexes is interesting, particularly the fact that downregulation of WASHC2C reduces viral titers (fig. 3E). Did the authors confirm that the cells are not affected in any other way upon siWASHC2C? If this treatment itself affects cell viability, it could also lead to lower viral titers. At least the authors should show that viral entry is not impacted, since this interaction seems to only occur at later time points of infection. It could also be interesting that the authors perform the TPCA experiment upon siWASHC2C +/- viral infection (although, I understand that this could be quite a large experiment).

5. As in my previous comment, the experiments with siCD63 (Fig 4h) need to have proper controls that this does not affect viability of cells. Perhaps the authors could perform the same microscopy experiment as in Fig 4e after siCD63 treatment and show that in that case ITGB1 is not internalized.

6. Related to the previous comment, the authors begin the next section with the following sentence (line 276): “Although our analysis based on $|\Delta T_m|$ captured numerous protein alterations, as exemplified by ECM components (Fig. 4c), it misses the subset of proteins that display non-sigmoidal melting curves.”, which I completely agree with. Although, when I looked at ITGB1 and ITGA2 melting curves, they themselves do not cross the 50% solubility threshold, how did the authors calculate ΔT_m for those? Also, are there any protein pairs in which the ΔT_m analysis is superior to the ΔAUC . The authors could show a correlation plot between these two metrics or alternatively simplify the manuscript by including just the AUC analysis.

7. The same comment as my comments 4 and 5 applies to the KO of IGF2R (Fig 6G).

8. For the IFIT1 IP (Figure 8E) the authors should also perform the experiment at 48h, if their hypothesis is that this protein is not interacting (or at least it is interacting less strongly) with pUL52 at that time point, since the melting curves do not overlap.

Minor comments:

9. Line 162: When CORUM is introduced, it is important to point out that the authors at this point are focusing solely on human complexes and disregarding the viral proteins.

10. Fig. 1E-F: the color coding of these panels is confusing. The authors should consider a different color scheme and keep it harmonized throughout the manuscript. In fig. 1f, when z-scores are

introduced, it would be helpful to guide the reader to what they actually mean (this comes in handy in fig 2A)

11. Fig 2A, the authors should clarify how many proteins/complexes are in each cluster. Is each line one complex or one protein? Also, the authors do not explain anywhere how the membership score (color coding for that figure) is calculated and what it means. In this figure, it could also be beneficial to keep the same y-axis scale for all clusters, to be able to compare effect size between clusters. For example, the authors focus on cluster #3 and #5, but these are complexes with very small changes.

12. Line 360: "One plausible explanation is that proteins with higher molecular weights tend to have lower T_m ." This has been previously observed and the authors should give proper credit to previous work.

13. Line 368: "Therefore, the difference in T_m values between viral and host proteins are primarily driven by molecular weights." The authors should tone down this claim, as the MW explains about 4% of T_m . There are many other factors, such as the structure of the protein itself that can alter the intrinsic thermal stability of proteins.

14. The rationale for switching from Euclidean distances to derivative of Euclidean distances (Ex) in the middle of the manuscript needs to be better explained. What is the advantage compared to Euclidean distances, or the z-values?

Sincerely,

Mikhail Savitski

We thank the editor and the reviewers for the careful assessment of our manuscript and for their insightful comments and recommendations. Based on the reviewer's comments, we have performed a series of additional experiments and have carefully revised the text to fully address all of the reviewer's concerns. Specifically, we have added controls that demonstrate that our siRNA-mediated knockdowns do not affect cell viability, additional reciprocal isolations to show differential interactions at distinct infection time points, new microscopy analyses, western blot analyses of viral proteins to determine impact on virus entry, and additional PRM analyses. The new experiments are in complete agreement with our initial conclusions. As a result, we have added 11 new figures or figure panels in the main manuscript (Fig. 3e, 3g, 4e, 4g, 5a, 6a, 6b, 6h, 8b, 8e, 9) and 9 new supplementary figures or figure panels (Supplementary Fig. 1b, 1d, 2, 3, 4, 5b, 6, 7, 9). Our point-by-point answer to the reviewers' comments follows with each reply marked by "**>Response:**".

Reviewers' comments:

Reviewer #1 (Remarks to the Author):

Hashimoto and colleagues had done a remarkable work deploying cellular thermal shift assay and quantitative mass spectrometry with the recent TPCA (thermal proximity co-aggregation) methodology to study the dynamics of protein complex in primary human fibroblast during HCMV infection. In addition to the approaches in Tan et al (2018), I am glad they had extended the methodology to include multiple time points and inter-species comparison that were accompanied by new analysis workflow. Their work and data provided a system view of proteins and interactions that are post-translationally modulated at different stage of HCMV infection. Quality of their work, data and analysis is exemplified by validation of multiple modulated complexes identified with Co-IP and microscopy (co-localisation), and the linkage of the associated proteins to virus production through siRNA.

The authors have also made conceptual advances showing that similar co-aggregation profiles of their data can be used to identify novel association among small group of viral proteins. For a large set that consist of both viral and human proteins, they demonstrated that their data can be used to construct a network of proteins with high co-aggregation similarity that was used with the COACH algorithm to identify potentially novel viral-host protein complexes. In particular, they experimentally validated novel interactions between viral protein pUL52 with IFIT1 and IFIT2 proteins inferred from such analysis. The manuscript is also very well-written and organised. What I think is lacking is a final figure depicting the virus infection process but highlighting the key findings or observation from this work. Otherwise, below are my comments and suggestions that could be useful for improving the work further.

>Response: We thank the reviewer for the constructive suggestions and comments. We agree that a final figure depicting the virus infection process and key findings from our work would help the reader. We have now added this as new Figure 9.

Major

1. The authors had highlighted that the T_m of a subset of proteins could not be computed as they

display non-sigmoidal curves, thus had used a complementary measure, difference in area under the curve (AUC), to identify more differentially modulated proteins. Nevertheless, it is possible that the shape of a curve could change with minimum difference in AUC. Euclidean distance between the curves of the same protein at different time point could be used to detect such changes although the measure lacks information on the direction of change. I will appreciate if the authors could do a comparative analysis whether there is substantial difference between AUC or Euclidean distance for identifying differentially modulated proteins in HCMV infection.

>Response: We thank the reviewer for raising this question, which led us to realize that we did not previously properly define how ΔAUC was calculated in our manuscript. We now provide a description of Euclidean distance and ΔAUC calculations in our study.

The Euclidean distance, as in prior reports, is defined as $d(p,q) = \sqrt{\sum_{T \in \mathbb{T}} (p_T - q_T)^2}$, where p_T and q_T represent normalized adjusted solubility of proteins p and q at heating temperature T , respectively. We employed this calculation to assess the solubility differences between different proteins at one given time point.

The ΔAUC is actually based on the Euclidean distance calculation proposed by the reviewer,

being calculated as $\Delta\text{AUC}(t) = \sqrt{\sum_{T \in \mathbb{T}} (p_{t,T} - p_{mock,T})^2}$, where T represents the heating temperature, and p_t and p_{mock} represent the normalized adjusted solubility of protein p at an infection time point t and uninfected state *mock*, respectively. Therefore, both measurements will generate positive values, and are not sensitive to the direction of the changes. We are sorry that our nomenclature was confusing. We used it to differentiate this measurement from the Euclidean distance calculation mentioned above, as in this case we assessed the change of solubility of a given protein across infection time points. Additionally, we used this terminology given that our initial comparison of this calculation and the absolute area under the curve provided analogous information (see Supplementary Fig. 6), and it was easier to explain to a reader the type of information obtained. To fully address this question, we have now changed the terminology and have added detailed explanations. Specifically:

- a. We revised the Figure 5a panel to better illustrate the calculation.
- b. We changed the terminology from ΔAUC to temporal Euclidean distance (TED).
- c. We added the description and equations for all these calculations to the methods section on pg. 45-46.
- d. We included the new Supplementary Fig. 6, to illustrate to a reader that this temporal Euclidean distance calculation provides information similar to that obtained from measuring the absolute area under the curve.

2. On the observation that most protein known to or potentially translocate do not exhibit much difference in T_m values, is the observation similar based on AUC or euclidean distance? How much is known about the portion of these proteins that translocate? If only a small portion translocate, the non-translocating subset will dominate the signal. In addition, for known translocating proteins with observable difference in T_m , are there enrichment in any organelle or cellular location?

>Response: The reviewer raises several good questions, and we performed additional analyses to address these.

We now compared the distribution of the derivative of TED (prior AUC) for non-translocating proteins and translocating proteins (predicted by Jean Beltran et al. and the translocatome by Mendik et al.), and the results are shown in new Fig. 6b and Supplementary Fig. 9b. The majority of the proteins predicted to translocate did not display a change in aggregation profiles.

Regarding the reviewer's comment that the non-translocating subset of proteins dominates the dataset, this is true. Specifically, 5% of our detected proteins have been predicted to translocate specifically during HCMV infection (based on Jean Beltran et al.), and 12% of the proteins are broadly predicted to translocate (Mendik et al., translocatome). However, our calculations showing the distributions of the derivative of TED now separate the non-translocating from the translocating proteins.

With regarding to enrichment within translocating proteins of a certain subcellular localization, we now show that the subset of translocating proteins that displayed a change in thermal stability was mainly contributed to by ribosomal proteins (Fig. 6b, compare orange to green lines; Supplementary Fig. 9b).

Finally, we also included a comparison between ΔT_m and TED, and observed that these were positively correlated (albeit a weak correlation). Overall, this suggests that both of these calculations lead to similar conclusions (Fig. 6a, Supplementary Fig. 9a). This comparison also supported the notion that TED may better capture the thermal shift for proteins whose thermal profiles cannot be fitted to a sigmoidal curve. This is especially the case for proteins with T_m values greater than 65°C, which are reflected in the data points that fall at $-10 \log_e \Delta T_m$.

We now include information regarding these results and analyses in the results section (pg. 21).

Minor

3. Line 107: As it not clear that proteins in the same complex need to unfold similarly (it may be well possible a subunit could unfold first to induce precipitation) to have similar (in)solubility, perhaps the sentence could be edited to "This method is based on the principle that proteins in the same complex exhibit similar progressive insolubility when treated with an increasing temperature gradient"

>Response: We agree and have modified the text accordingly on pg. 7.

4. Table S1C: Are the values in the table MS2 intensity of reporter ions? If so, the use of word "Abundances" in description (sheet 1) is potentially misleading.

>Response: We apologize that this phrasing was unclear. We used the word "abundances" to indicate that these values are the sum of the peptide reporter ion intensities for all peptides shared across time points for each indicated protein. To further clarify this, we have added this description in the methods section (p. 46-47) and have modified the wording on sheet 1 for Table S1C.

5. Line 148: Please kindly clarify how the correlation coefficients (for reproducibility analysis) are computed. Are solubility ratios or raw reporter intensities used?

>Response: We thank the reviewer for pointing out that this was not clear. We have now added a sentence regarding this in the methods section. Specifically, the correlation coefficients were calculated based on the \log_e transformed abundances, estimated from the sum of the peptide reporter ion intensities for peptides shared across time points for the indicated proteins. This information was added on pg. 46-47, as well as in the corresponding figure legends in Figure 1b and Supplementary Fig. 1c.

6. Supplementary Fig 1b. It is unclear what are the axis of individual scatterplot: are they \log_{10} or \log_2 of reporter ion intensity or something else?

>Response: In Supplementary Fig. 1b, the scatterplots represent the \log_e -transformed abundances (i.e., the sum of the peptide reporter ion intensities for indicated proteins). We have now added this information to the figure legend.

7. Line 151: Fig 1c depicts increasing number of viral protein being identified as infection progresses that may not necessary suggest increased abundance of individual viral proteins. For viral proteins identified across all time points (except mock, of course), maybe the changes in abundance of each protein can be expressed as fraction at 24hrs and the data distribution visualised as a box plot.

>Response: We agree with the reviewer that plotting the change in abundance of viral proteins compared to 24 hpi is helpful for visualizing the increased abundance of viral proteins. As the reviewer suggested, we made a box and whisker plot to show the fold change of viral proteins (relative to 24 hpi) that were quantified across all time points during infection. Viral protein abundances increase as infection progresses compared to 24hpi, and this observation is consistent between replicates. This graph has been added as new Supplementary Fig. 1b.

8. Fig 1f: Shouldn't the legend be

Z score:>2.0

Z score:1.5 to 2.0

Z score:1 to 1.5

>Response: The reviewer is correct, and we have made this change in the Fig. 1f legend.

9. Line 151-154: Please kindly clarify whether abundance ratio or raw reporter intensity is used?

>Response: Thank you for the comment. For plotting the proteome abundance data in PCA space, we used abundances calculated as the sum of all peptide reporter ion intensities for each protein. We now clarify this in the methods section on pg. 47, as well as in the corresponding figure legend (Figure 1d).

10. Fig 3b. Maybe the subunits(node) of CCC complex and WASH complex can be uniquely coloured.

>Response: We appreciate the reviewer's suggestion, and have now revised the figure to better differentiate the colors of the WASHC2C (member of the WASH complex) and the CCC complex. To keep the figure consistent, we have selected these colors to match the color in the model depicted in Fig. 3g.

11. Fig 8d: Might be helpful to explicitly indicate on the diagram or caption that pUL52 is not detected in Mock and at 24hpi.

>Response: We thank the reviewer for the advice, as this makes the figure clearer. We have now indicated on the graphs in Fig. 8d that pUL52 is not detected in mock and 24 hpi, and we have also included this information in the figure legend.

12. Line 771-773: It is unclear why for the GO enrichment analysis, the minimum number of complexes per GO term need to be set at 6? If the set is small, signal will not be significant more often than not. Is this for reason of multiple hypothesis correction?

>Response: This original threshold was primarily selected to reduce the redundancy of the GO terms and focus on the main biological processes to help with the clarity of the figure. The enrichment scores were based on the p-values of the Fisher's exact test, being calculated as $-\log_{10} p$ -value. The threshold of the minimum number of complexes per GO term was set after we calculated the enrichment scores, so changing this threshold would not be expected to impact the enrichment scores. Therefore, there is no difference between the enrichment scores associated with any cutoff of protein complexes per term. However, to fully address this comment, we have revised the Supplementary fig. 2 to now list the enriched GO terms without a limit to the number of complexes assigned to a GO term. We have expanded our method section to explain this (pg. 48-49).

Additionally, we have further revised Fig. 2 and its legend to increase its clarity and the level of information provided. Specifically, we have added the number of complexes per cluster (Fig. 2a), have indicated in the legend that each line in the cluster represents one protein complex, and have revised the y-axes so that all clusters are shown on the same scale. To further improve the visualization of the different trends, we have increased the number of clusters to 8, which allowed us to better highlight the terms associated with specific profiles. Finally, we have included a detailed description of the membership score calculation to the methods and partly in the figure legend.

13. Table 2A: The subunits of each protein complexes with curves should be listed.

>Response: We have now listed the subunits of each complex in Table S2A and B.

Chris Soon Heng Tan

Reviewer #2 (Remarks to the Author):

The manuscript entitled “Temporal dynamics of protein complex formation and dissociation during human cytomegalovirus infection” by Hashimoto et al utilizes a thermal shift assay paired with mass spectroscopy to define the cellular protein interactions that occur during a viral life cycle. The strategy is quite innovative and has the potential to reveal important protein interactions that are critical for virus infection, replication, and dissemination. The manuscript describes several protein interactions CCC-WASH complex and ITGB1-CD63 as well as other cellular factors that have been demonstrated to play a role in virus infection. The findings were also validated by siRNA data studies to demonstrate the role of the identified protein. Even though most of the protein identified in this study have been shown to play a role in CMV infection in previous studies, the strategy to identify these protein interactions and validation of the findings is quite novel. However, there are some technical issues that would enhance the manuscript for a reader not well versed in mass spectroscopy.

1) The analysis of protein complexes should be better described in the Results Section. A diagram of how protein-protein complexes were identified should be included.

>Response: We appreciate the reviewer’s suggestion, and have now included a new figure panel illustrating the workflow for the analysis of putative protein complexes based on the CORUM database (new Supplementary Figure S1c). We also provided a description of this analysis in our manuscript. Briefly, proteins quantified in our datasets were searched in the CORUM database, which contains curated human protein-protein complexes that have prior experimental validation from the literature. This information regarding putative complexes present in our dataset was used to determine whether the complex members had similar aggregation profiles in fibroblasts, suggesting their possible association. For this, average Euclidean distances were calculated between the protein aggregation curves of the members of predicted complexes. This calculation was repeated at each infection time point to determine whether these putative associations change during the progression of infection. For certain complexes and protein-protein interactions of interest, we further followed these association predictions with validation and functional follow-up analyses. We include this information in the results and methods sections (pg. 10 and 49-50), and refer to the new figure panel.

2) The manuscript should better describe the relevance of Euclidean distances and how they can be derived from the mass spec data. In fact, the Average Euclidean distances in Figure 1e and CORUM complexes in Figure 1f do not show significance differences. Is this typical data?

>Response: We thank the reviewer for highlighting the need to better describe the Euclidean distances. We have expanded our methods section to explain the different types of Euclidean distance calculations that we have performed, and the specific information obtained from each of these measurements (pg. 45-46).

Specifically, we calculated Euclidean distance to assess the solubility differences between different proteins at one given time point. Euclidean distance $d(p,q)$ is defined as =

$\sqrt{\sum_{T \in \mathbb{T}} (p_T - q_T)^2}$, where p_T and q_T represent normalized adjusted solubility of proteins p and q at heating temperature T .

Additionally, to assess the change in the solubility of a given protein across infection time points, we measured temporal Euclidean distances calculated as $\Delta AUC(t) =$

$\sqrt{\sum_{T \in \mathbb{T}} (p_{t,T} - p_{mock,T})^2}$, where T represents the heating temperature, and p_t and p_{mock} represent the normalized adjusted solubility of protein p at an infection time point t and uninfected state $mock$, respectively. We now reworded this calculation as temporal Euclidean distance (TED) to clarify that this measurement is still based on the Euclidean distance, but for one given protein across infection time points.

To assess the putative formation or disruption of a protein complex, we measured average Euclidean distances for members of protein complexes. The average Euclidean distance was defined as $\frac{\sum_{p,q \in c} d(p,q)}{\binom{|c|}{2}}$, where c , $|c|$, and $d(p,q)$ represent proteins identified as members of protein complexes in both CORUM and our TMT datasets, the number of proteins in c , and the distance between proteins p and q, respectively. The average Euclidean distances were compared for CORUM complexes and random proteins (i.e., containing the same number of randomly selected proteins as those identified as putative CORUM complex members from our dataset). To better display the data, we also utilized the derivative Euclidean distance (Ex) ($Ex = 1/(1+E)$) which constrains the data range to (0, 1). Additionally, this transformation leads to a better fitting to a normal distribution compared to Euclidean distance. We have now clarified these measurements and their specific benefits in the methods section (pg. 45-46).

It is typical for the differences between average Euclidean distances to not be very large and for the ranges of average Euclidean distances to overlap. These results are in line with previously published data. For example, in the elegant thermal profiling paper from Tan et al. (*Science*, 2018), the distribution of average Euclidean distances of CORUM data was 0.28 to 0.82, and the distribution of average Euclidean distances of random data was 0.4 to 0.9. Our data displays similar trends, with the distribution of average Euclidean distances of CORUM data being 0.4 to 1, and the distribution of average Euclidean distances of random data being 0.7 to 1.3. However, when a Mann-Whitney U test is performed, there is a statistically significant difference between the average Euclidean distance for CORUM complexes and the average Euclidean distance for random proteins (Fig. 1e).

3) All of the bar graphs should include a statistical analysis.

>Response: As the reviewer has suggested, we have now included stars indicating the p-value significance for the relevant bar graphs with three or more replicates (Fig. 3e, 3f, 4g, 4h, 5h, 6g, 6h). We have also shown all data points on the bar graphs.

4) As represented, Figure 3b is very confusing. The Figure should be modified.

>Response: We appreciate the reviewer's suggestion, and have now revised Fig. 3b to improve its clarity. First, we have adjusted the color scheme of the lines between proteins to better

visualize the differences in the z-scores. As a reminder, the low z-score suggests a weak association (lines with blue tones), while a high z-score suggests a stronger association (lines with orange tones) between proteins. We have now also added this information in the legend of Figure 3. Second, as each node represents a protein that is either a member of the WASH or CCC complex, we have colored the protein nodes to match the colors used to depict these complexes in the model in Fig. 3g. Third, to further delineate the two complexes we have circled the CCC complex with a dashed line and labelled both the members of the CCC and WASH complexes on the figure. These changes should facilitate comparing the z-scores for protein pairs in uninfected cells (mock) and z-scores for protein pairs in cells at 72 hours post infection (hpi). The adjusted color scheme highlights that the results from thermal profiling suggest an increased association between WASHC2C, a core component of the WASH complex, and members of the CCC complex (lines changing from blue in mock sample to orange at 72 hpi).

5) Figure 3c IP-PRM analysis is not very convincing. Can this be improved?

>Response: This was a challenging experiment to perform given several technical constraints. Our thermal profiling data suggested the increased association of the endogenous forms of WASHC2C and members of the CCC complex late in HCMV infection. However, given that antibodies against WASHC2C that are suitable for immunoaffinity purification are not available, we constructed FLAG-tagged WASHC2C. This construct was transfected into fibroblasts, and immunoaffinity purifications were performed in uninfected and infected cells. One caveat of this approach is that the tagged version of the protein and the endogenous protein are present at the same time. Therefore, the endogenous members of the CCC complex will associate with both of these forms of WASHC2C, possibly even having a preference for the endogenous WASHC2C over the tagged version. Given this limitation of the approach, the fact that we can still partly recapitulate the increased association of WASHC2C with CCC following infection is quite encouraging. We have now added additional sentences in our results section to clarify the approach used (pg. 13).

Additionally, to further support our findings of a role for WASHC2C in virus production, we have also included a new panel in this figure to demonstrate that the knockdown of WASHC2C does not affect cell viability (new Figure panel 3e).

6) The viral titer validation assays are a little confusing because the inclusion of siRNA would knock-down the gene for several days, but a virus titer assay is typically 10-14 days. This should be clarified.

>Response: We thank the reviewer for the comment, which we have addressed with both text revision and new experiments. In fact, this is an aspect that we have carefully considered and optimized from the beginning. Specifically, we have previously optimized this knockdown protocol in the context of HCMV infection to ensure the maximal knockdown of a targeted protein throughout the entire viral replication cycle, and we and others have reported such optimizations in other studies (e.g., Koyunku et al. mBio 2014 PMID:25516616, Lee et al. mBio 2019 - PMID:31594813). However, based on the reviewer's comment, we realize the need to clarify how the viral titer assay was performed in the current manuscript. Briefly, cells treated

with siRNAs were infected with HCMV two days post siRNA transfection and re-treated with the siRNAs on the same day of infection to ensure that the genes of interest stayed knocked down throughout the viral replication cycle.

Regarding the measurements of virus titers, given that HCMV has a five day replication cycle, the supernatant (representing released viral particles) was collected from the infected cells 5 days after infection. This collected supernatant was used to infect a reporter plate of wild-type fibroblasts. At 24 hpi, the cells on the reporter plate were fixed and stained with an antibody that recognizes an immediate early viral protein. The number of infected cells could then be used to determine the infectious units/mL (viral titers) produced from cells the siRNA-treated cells for each of the siRNA conditions. This method for assessing virus titers is well documented in the literature, being used by different virology groups, including ours.

We have now added information about the siRNA treatment and the viral titer assay to the methods (pg. 56-57).

Additionally, we have performed new experiments to validate the maintenance of the siRNA-mediated knockdowns at late stages of HCMV infection. Specifically, we performed PRM analyses to demonstrate that effective knockdowns are still observed at 72 hpi and 120 hpi (new Supplementary Fig. 4).

Reviewer #3 (Remarks to the Author):

Comments to authors:

In this work, Hashimoto et al. generate a temporally-resolved dataset of protein thermal stability and abundance during viral infection. This resource on its own is potentially valuable for the research community and might support hypotheses from other research groups. The authors go through some length to try to link some of the observations to known biology and potentially find hints of new molecular targets to fight infection. However, I think some of those experiments lack proper controls. Overall, the manuscript is well written and generally clear in its message, although some of the concepts/terms used might not be easily understood by a broad audience and should be more clear. I offer some comments that I think would greatly improve the manuscript.

>Response: We thank the reviewer for all the constructive comments. We have performed all the suggested experiments, and have added new figure panels with controls and additional IPs. We have also revised the text to include additional descriptions and discussion, as suggested.

Major comments:

1. As a general comment, the authors focus most of the manuscript on protein-protein interactions. While this is interesting, there are many reasons for changes in T_m of proteins that are unrelated to direct protein interactions (e.g., binding of small molecules (metabolites), binding to nucleic acids, changes in protein localization, PTMs). Some of the apparent reduced/increased distance between protein pairs might be just by chance, because one of the proteins is engaged in one of the above mentioned interactions. When I looked into some melting curves of protein pairs discussed in the manuscript, many times the pairs that got closer do not always completely overlap, as would be expected if they are co-aggregating. Also, some times the pair got closer by the thermal destabilization of one of the proteins. I have trouble

rationalizing why this would happen, i.e., if a protein is more thermostable when alone, why would it become less thermostable when engaged in an interaction.

The authors should be more careful in their overinterpretation of the data and as a minimum inform the reader that some of the changes might not be related to PPIs. Thermal proteome profiling has been shown to inform on other processes such as e.g. protein metabolite interactions and pathway activity (PMID: 30858367, PMID: 29980614, PMID: 29706546). The possibility of such interpretations should be taken into account and at the very least mentioned. I would suggest that the authors look more carefully into the function of the proteins discussed to ensure that neither of the other mechanisms is likely to cause the changes in thermal stability. It would also be helpful to have supplementary figures with the melting curves of the pairs of proteins described.

>Response: We thank the reviewer for making this thoughtful comment, which highlights both the power of this approach, as well the challenges associated with interpreting the rich data obtained. It is evident that this method provides valuable information when a biological event leads to alterations in protein properties. This can be seen as a strength, as well as an interpretation challenge, that this information can point to various biological factors that may induce these alterations, such as changes in protein localization, protein interactions with different types of molecules, and PTMs. We completely agree with the reviewer's comment, and fully followed all the provided suggestions. The reviewer is correct that association between a pair of proteins would usually suggest increased thermal stability. During viral infection, global alterations are known to occur in protein localizations, as well as the levels of viral proteins, protein modifications, metabolites, and small molecules. As the reviewer notes, these changes may contribute to protein complex formation, and could alter the polarity of the interface between the protein complexes and the surrounding solvent. Therefore, it is possible that some factors induced by HCMV infection would allow for protein coaggregation while the solubility of one component may be reduced. We have now added sentences in the results section to indicate that a range of factors can influence protein T_m values, and have added additional citations, as requested (pg. 9-10).

With regard to the specific protein examples that we focus on in our paper, we have now expanded the discussion to place our results in the broader context of knowledge regarding changes in localization and PTMs, and how these may also contribute to T_m. Below we list some examples.

For the protein pairs that we highlight in this study (ITGB1/CD63, WASHC2C/CCC complex, and IFIT1/IFIT2/pUL52), only CD63 and ITGB1 have well-documented, functional PTMs. CD63, like all tetraspanin proteins, is heavily glycosylated (Yunta and Lazo Cell Signal. 2003, PMID: 12681443). ITGB1 is known to be acetylated and ubiquitinated. Functionally, it has been shown that ITGB1 can be ubiquitinated at K794, and ubiquitination of ITGB1 during HCMV infection facilitates its degradation (Lobert et al. Dev. Cell 2010, PMID: 20643357). Additionally, in a previous study from our lab, we detected acetylation of this same lysine residue during HCMV infection (Murray, et al. Nat Commun. 2018, PMID: 30470744), suggesting that these PTMs may act as a switch on/off mechanism that either promotes or prevents ITGB1 degradation. While it is possible that the PTM modification of CD63 and/or ITGB1 may contribute to the alterations in their T_m values during infection, multiple publications have determined that these two proteins interact when CD63 is recruited to the plasma membrane (Lee et al. Biochem. J 2014, PMID: 24635319, Jung et al. EMBO J. 2006,

PMID: 16917503). These findings support our hypothesis that the alterations in T_m that we observe for CD63 and ITGB1 are most likely driven by infection-induced changes in CD63 localization that facilitate an interaction between these two proteins initially at the plasma membrane, followed by their internalization. It is possible that the PTM modulation acts downstream of this change in localization to promote ITGB1 degradation following its internalization. We have added these concepts to the discussion section (pg. 32).

As the reviewer points out, changes in protein localization can contribute to changes in protein T_m values. The example discussed above points to such a case. Another example is provided by our identification of changes in the T_m of the translocating protein IGF2R. Similarly, it has been shown that the WASH complex can recruit the CCC complex to facilitate trafficking of ATP7A and LDLR (Phillips-Krawczak et al. Mol. Biol. Cell 2015, PMID: 25355947, Bartuzi et al. Nat. Commun. 2016 PMID: 26965651). Therefore, we propose that the WASH complex recruits the CCC complex during HCMV infection to facilitate different trafficking patterns. One possible explanation for the incomplete overlap between the WASHC2C and CCC complex member T_m curves is the likely presence of different pools of these proteins that do not form a WASHC2C/CCC joint complex and therefore are likely to have different T_m values. We initially mentioned this limitation of this approach in our discussion, indicating the challenge with analyzing proteins that may be part of multiple complexes. We have now expanded this part of the discussion, to specifically indicate that this can impact T_m values and melting curves (pg. 36).

Additionally, as the reviewer indicates, protein association with metabolites can also impact T_m values, as recently reported for ATP and GTP (Sridharan, et al. Nat. Commun. 2019 PMID: 30858367). It has been shown that HCMV infection does not alter AMP, ADP, or ATP levels (Munger, et al. PLoS Pathog. 2006, PMID: 17173481). Additionally, none of the protein pairs that we focused on are known to bind ATP or GTP. However, the reviewer's point is well taken, as HCMV infection is known to alter cellular metabolism, in particular by inducing glycolytic flux. Therefore, it is possible that changes in associations to metabolites may contribute to alterations in melting curves. We have now included this concept and references in the results section that describes our findings for metabolic proteins (pg. 19), as well as at the end of the first results section that presents the approach (pg. 9-10).

Finally, as per the suggestion of the reviewer, we have now included all the melting curves over the time course of infection for the protein pairs that we highlight (PI3K proteins, WASHC2C/CCC complex, and ITGB1/CD63) in Supplementary Fig. 3. The melting curves for IFIT1/IFIT2/pUL52 can be found in Figure 8.

2. Related to the previous comment, although it would require major rearrangement of the manuscript, I think a more logical way to tell this story would be to look first at ΔT_m or ΔAUC throughout infection and which proteins/processes are changes. Then, try to rationalize some of these changes, which in part would reflect the spectrum of PPIs changing, but also many other types of interactions/biophysical changes to the protein. Related to that, the original name of the combination of CETSA with quantitative mass spectrometry, "thermal proteome profiling, TPP", should be mentioned, since this is what is used in the manuscript.

>Response: We thank the reviewer for the comment, and we agree that it is important to mention “thermal proteome profiling (TPP)”, and we have now included references to TPP in different parts of introduction, results and discussion sections (pg. 6, 7, 8, and 30).

We appreciate the reviewer’s suggestion that the manuscript may flow better with a different organization. However, when focusing on protein-protein interactions and considering the complexity of the data, a relatively simpler starting point is to specifically analyze members of known protein complexes. As the reviewer nicely pointed out in the first comment, a wide range of factors can contribute to a protein’s T_m . Therefore, the temporal changes in the melting curve of a given protein may not specifically indicate a change in its associations with other proteins, and the results are more challenging to interpret. However, by starting with prior knowledge regarding functional complexes, the initial analysis is more focused, allowing us to ask more specific questions. In particular, this analysis provides information regarding the possible maintenance, formation or dissociation of functional complexes during infection. To perform this initial analysis of known protein complexes, we opted to measure Euclidean distances for the known members of complexes. If members of protein complexes have different solubility curves but the same T_m , this information would be insufficient to determine likely associations or dissociations within protein complexes. Euclidean distance helps to address this problem by using 10 dimensional data while T_m uses one dimensional data. Furthermore, since T_m is a physical property, fitting is required to interpret this data. The fitted Euclidean distances are less sensitive to noise than the T_m values

To further clarify this analysis and the value of this starting point, we now include a new figure panel illustrating the workflow for the analysis of putative protein complexes based on the CORUM database (new Supplementary Figure S1c). We also provided a description of this analysis in the results and methods sections (pg. 10 and 46). Overall, this initial analysis of protein complexes acted as both a confirmation of the approach in this context of viral infection (e.g., identification of expected infection-induced changes in known associations) and a discovery tool for new biology. Therefore, the success of this analysis gave us the confidence to then expand the analysis to other putative novel host-host and viral-host protein associations, and then understand some of the challenges with ΔT_m measurements and the usefulness of measuring temporal Euclidean distances.

We have also expanded our methods section to explain the different types of Euclidean distance calculations that we have performed, and the specific information obtained from each of these measurements (pg. 45-46).

Additionally, we have edited transition sentences between subsections to improve the intended flow of the manuscript.

3. Throughout the text, the authors compare all their results to a “mock” infection. I could not find in the methods section what this means (from methods section: line 597 “The media was then replaced with fresh culture medium, and infected cells were cultured for the indicated length of time.”). What is the time for the mock infection? For me, a better and important control would be a 0h time point (i.e., cells would go through the 2h initial infection and then medium change). If this is too long and too many things already happen in that time period, at least the cells should go through the same medium exchange that the infected cells go through, but without the virus.

>Response: We apologize for not being clear. The reviewer is correct that, in order to provide an appropriate control, the uninfected cells (mock) have to go through the same medium exchange as the infected cells. The “mock” terminology is a classical virology description of exactly this type of control uninfected cells, which have to go through the same processing steps as the infected cells, but are not exposed to the virus. To briefly explain the routine process, in order to promote virus entry, cells are initially infected in half volume media. The virus is allowed to enter for 2 hours, and then the media is changed with full volume, fresh media. Therefore, the control cells for this experiment are uninfected cells that have been subjected to the same media treatment, i.e., cultured for 2 hours in half volume media, followed by exchange of media with full volume, fresh media. This is why the virology community refers to these cells as “mock”-infected, and, in agreement with the reviewer’s comment, considers these the appropriate control cells. The text in the methods section titled “Cell culture and viral infection” has been modified to more clearly explain how the control mock cells were prepared (pg. 37).

4. The results of the interaction between the WASH and CCC complexes is interesting, particularly the fact that downregulation of WASHC2C reduces viral titers (fig. 3E). Did the authors confirm that the cells are not affected in any other way upon siWASHC2C? If this treatment itself affects cell viability, it could also lead to lower viral titers. At least the authors should show that viral entry is not impacted, since this interaction seems to only occur at later time points of infection. It could also be interesting that the authors perform the TPCA experiment upon siWASHC2C +/- viral infection (although, I understand that this could be quite a large experiment).

>Response: We appreciate the reviewer’s comments, and have now performed additional experiments as suggested. We understand the concern that the changes in viral titers from WASHC2C knockdown cells could result from knockdown induced cell death. To address this possibility, we performed a TUNEL cell death assay to assess how this siRNA treatment affects cell viability. In this assay, we included a negative control which was not treated with TUNEL labelling reagent, a positive control treated with DNase I to induce DNA damage, siGFP as a negative control for siRNA treatment, and siWASHC2C constructs 1 and 2. Based on the results, we observed that siRNA treatments in our study did not induce cell death. Therefore, these results support that the reduced viral titers upon WASHC2C knockdown are not a reflection of compromised cell viability, but rather an effect on infectious virus production. We now include this data in Figure 3E. Furthermore, we also include additional analyses that confirm that the WASHC2C knockdown is maintained throughout late stages of HCMV infection (new Supplementary Fig. 4).

Additionally, as the reviewer suggested, we assessed the effect of WASHC2C knockdown on viral entry and virus protein production by performing a western blot analysis in siWASHC2C and siControl cells across the infection time course (uninfected, 24, 48, 72, 96, 120hpi). We monitored the levels of representative immediate early (IE1), delayed early (pUL26), and late (pUL99) viral proteins. As we expected, siWASHC2C cells displayed similar levels of viral proteins to siControl cells (Figure 3g). IE1 levels at 24 hpi are similar, indicating that WASHC2C is not affecting viral entry. Additionally, pUL26 and pUL99 levels are also not decreased, in agreement with WASHC2C having a role at later stages of infection (after viral protein production), likely in intracellular trafficking events. There might be a slight increase in

the levels of pUL26 and pUL99 at 96 and 120 hpi in siWASHC2C cells, which may suggest a slight accumulation of late viral proteins, possibly due to an inability for viral particles to effectively be trafficked during their assembly or egress from the cell. We have now included the description of these new results on pg. 14-15.

We agree with the reviewer that a TPCA experiment on siWASHC2C +/- cells during viral infection would be useful for better understanding the role of this protein during viral infection. However, we feel that this experiment, which the reviewer acknowledges would be quite large, is outside the scope of this study.

5. As in my previous comment, the experiments with siCD63 (Fig 4h) need to have proper controls that this does not affect viability of cells. Perhaps the authors could perform the same microscopy experiment as in Fig 4e after siCD63 treatment and show that in that case ITGB1 is not internalized.

>Response: We agree with the reviewer's comments, and have performed all the suggested experiments. We have performed a TUNEL cell death assay following CD63 knockdown, demonstrating that this knockdown does not affect cell viability (new Figure 4g). The experimental approach (positive and negative controls) was similar to that described in the point above. Furthermore, we include additional analyses that confirm that the CD63 knockdown is maintained throughout late stages of HCMV infection (new Supplementary Fig. 4).

As the reviewer suggested, we also performed microscopy analyses following CD63 knockdown. In agreement with our initial hypothesis, we observed more ITGB1 retained at the plasma membrane at 96 hpi in CD63-knockdown cells in comparison to control cells (new Fig. 4e). We have now included these results on pg. 17.

6. Related to the previous comment, the authors begin the next section with the following sentence (line 276): "Although our analysis based on $|\Delta T_m|$ captured numerous protein alterations, as exemplified by ECM components (Fig. 4c), it misses the subset of proteins that display non-sigmoidal melting curves.", which I completely agree with. Although, when I looked at ITGB1 and ITGA2 melting curves, they themselves do not cross the 50% solubility threshold, how did the authors calculate ΔT_m for those? Also, are there any protein pairs in which the ΔT_m analysis is superior to the ΔAUC . The authors could show a correlation plot between these two metrics or alternatively simplify the manuscript by including just the AUC analysis.

>Response: To address the reviewer's comment, we have edited the text and added additional supplementary figures to explain the procedure of curve fitting and the comparison of $|\Delta T_m|$ and ΔAUC . The melting curve of each protein was fitted to the log-logistic function, $LL(t) = c + \frac{1-c}{1+e^{b(\log t - \log a)}}$, which was also used in Tan et al. (2018), by using `optimize.curve_fit` from Scipy python package. As the reviewer points out, for some of the proteins, such as ITGB1 and ITGA2, the relative solubility of the identified proteins did not cross 50% across the 10 heating temperatures used. This is likely because these proteins were thermally stable within the range of these heating temperatures (37°C ~64°C). We used T_m as the fitted inflection point, and therefore, for some of these proteins, this value will not accurately represent the T_m . However,

once the thermal stability is changed during infection, the fitted inflection points is also likely to change, and $|\Delta T_m|$ based on the fitted inflection points would reflect an altered thermal stability.

In our study, $|\Delta T_m|$ values were utilized to perform an enrichment analysis of KEGG pathways, and the proteins involved in ECM_RECEPTOR_INTERACTION were most enriched. To extend the scope of our analysis, we included all quantified proteins associated with this term, which contained a few proteins without $|\Delta T_m|$ values, such as ITGA2. In Fig. 4c, we presented the TPCA-derived z-scores generated from Euclidean distances to examine predicted protein associations and dissociations within these pathways. Since the calculation of Euclidean distance does not depend on sigmoidal curve fitting, this measurement is more inclusive than $|\Delta T_m|$.

$|\Delta T_m|$ was calculated as the difference between the melting temperatures at two different time points, which indicates the shift of free energy at the midpoint of folded and unfolded states. Since this is a representation of the physical properties of proteins, we want to report these values for transparency and possible use by others (especially for the values in the uninfected samples). ΔAUC was calculated as $\Delta AUC(t) = \sqrt{\sum_{T \in \mathbb{T}} (p_{t,T} - p_{mock,T})^2}$, which is more sensitive to changes in the shape of curves. Therefore, this measurement assesses the change of solubility of a given protein across infection time points. Even though we could observe more changes using this measurement (as shown in Figure 5c), the ΔAUC values are specific to our study on infection time points, and may not be directly transferable to other studies outside of this biological context.

The $|\Delta T_m|$ and ΔAUC values are correlated in our study, and we have included graphs to compare these two measurements as follows (see new Supplementary Figure S7).

Lastly, we want to note that since, as shown above, the ΔAUC is based on the Euclidean distance calculation, for clarity we now replaced this AUC terminology with temporal Euclidean distance (TED). We initially used the AUC terminology to differentiate this measurement from the Euclidean distance calculation between different proteins at one given time point. Additionally, we used this terminology given that our initial comparison of this calculation and the absolute area under the curve provided analogous information (see new Supplementary Figure S6), and it was easier to explain to a reader the type of information obtained. However, we have now changed the terminology and have added detailed explanations of each calculation and the information obtained from each (in results on pg. 18 and in the method section on pg. 45-46).

7. The same comment as my comments 4 and 5 applies to the KO of IGF2R (Fig 6G).

>Response: We agree with the reviewer, and have now included the adequate control experiment to demonstrate that knockouts of IGF2R do not induce cell death (new Fig. 6h). For this, cell viability was evaluated by TUNEL cell death analysis, as described in responses to comments 4 and 5.

8. For the IFIT1 IP (Figure 8E) the authors should also perform the experiment at 48h, if their hypothesis is that this protein is not interacting (or at least it is interacting less strongly) with pUL52 at that time point, since the melting curves do not overlap.

>Response: We followed the reviewer's suggestion and have now performed an additional isolation of endogenous IFIT1 at 48 hpi (revised Figure 8e). In agreement with the thermal co-aggregation profiles, the IFIT1 IP led to the effective co-isolation of IFIT2 at both 48 hpi and 96 hpi, while the level of associated pUL52 was less pronounced at 48 hpi compared to 96 hpi. Therefore, these new results support our initial conclusion that pUL52 associates with IFIT1 and IFIT2 proteins predominantly at late stages of infection.

Minor comments:

9. Line 162: When CORUM is introduced, it is important to point out that the authors at this point are focusing solely on human complexes and disregarding the viral proteins.

>Response: We agree with the reviewer that it is important to indicate that the CORUM complexes only refer to complexes composed of human proteins. We have modified the text to indicate this (pg. 10). Additionally, we have included a new figure panel to clarify the workflow for the analysis of protein complexes in CORUM (Supplementary Fig. 1d).

10. Fig. 1E-F: the color coding of these panels is confusing. The authors should consider a different color scheme and keep it harmonized throughout the manuscript. In fig. 1f, when z-scores are introduced, it would be helpful to guide the reader to what they actually mean (this comes in handy in fig 2A)

>Response: We have altered the color scheme in Fig. 1e and 1f so that it better depicts the different values. We have now included descriptions of these z-scores in the legends of Fig. 1f and 2a, as well as expanded our methods sections on pg. 49-50.

11. Fig 2A, the authors should clarify how many proteins/complexes are in each cluster. Is each line one complex or one protein? Also, the authors do not explain anywhere how the membership score (color coding for that figure) is calculated and what it means. In this figure, it could also be beneficial to keep the same y-axis scale for all clusters, to be able to compare effect size between clusters. For example, the authors focus on cluster #3 and #5, but these are complexes with very small changes.

>Response: As suggested, we have made changes to the figure and figure legend. Specifically, we have added the number of complexes per cluster (Fig. 2a). In each cluster, each line is one protein complex. To further improve the visualization of the different trends, we have now also increased the number of clusters to 8, which allowed us to better highlight the terms associated with specific profiles. We have now also revised the y-axes so that all clusters are shown on the same scale.

Finally, we also include a detailed description of the membership score to the methods and partly in the Fig. 2 legend. Specifically, membership scores were calculated as $\max_i(\mu_{ik})$ of the final μ_{ik} ($0 \leq i \leq c$, $0 \leq k \leq n$), where n and c represent the number of data sets (protein complexes) and clusters (c set to 8). The final fuzzy partition coefficient μ_{ik} , ranging from 0 to 1,

is calculated via iterative optimization, and it represents the harmonic mean of the relative distance between complex k and the center of cluster i with a weighting fuzziness parameter m' (m' set to 2). μ_{ik} depicts the chances of protein complex k being assigned to cluster i. Altogether, the membership scores inform us of the most likely cluster a protein complex k is assigned to, and the relative distance of this complex from the average (centroid) of the assigned cluster. Briefly, higher membership scores (dark red lines) indicate how close the given protein complex is to the average of its corresponding cluster, while lower membership scores indicate larger deviation of the complex from this average. This method is detailed in TJ Ross, 2010. Ross, Timothy J. *Fuzzy Logic With Engineering Applications*, 3rd ed. Wiley. 2010. ISBN 978-0-470-74376-8 pp 352-353, eq 10.28 - 10.35, and we are including a reference to this publication in the corresponding methods section (pg. 48).

12. Line 360: “One plausible explanation is that proteins with higher molecular weights tend to have lower T_m .” This has been previously observed and the authors should give proper credit to previous work.

>Response: We fully agree that this was previously observed, and this is what we actually meant, i.e., to refer to prior knowledge regarding what can impact T_m . The same is true for our statement regarding the known impact of hydrophobicity and charged residues. The references were somehow inadvertently missed, and we have now included the following citations (Mateus, et al. (2018) *Mol Syst Biol* and Volkening, et al. (2019) *Mol Cell Proteomics*) on pg. 23 to ensure that proper credit is given.

13. Line 368: “Therefore, the difference in T_m values between viral and host proteins are primarily driven by molecular weights.” The authors should tone down this claim, as the MW explains about 4% of T_m . There are many other factors, such as the structure of the protein itself that can alter the intrinsic thermal stability of proteins.

>Response: We completely agree with the reviewer. We have now reworded the sentence, and have included a note regarding the possible contribution of other factors. The modified text reads as: “Therefore, the difference in T_m values between viral and host proteins is in part driven by differences in molecular weight. Other factors, such as differences in the structures of host and viral proteins may further contribute to the differences in T_m values between these groups of proteins.” and can be found on pg. 24.

14. The rationale for switching from Euclidean distances to derivative of Euclidean distances (Ex) in the middle of the manuscript needs to be better explained. What is the advantage compared to Euclidean distances, or the z-values?

>Response: As the reviewer suggested, we have now included description of this calculation and its benefits. The Euclidean distance is defined as $d(p,q) = \sqrt{\sum_{T \in \mathbb{T}} (p_T - q_T)^2}$, ranging from 0 to infinity, and its distribution does not follow a normal distribution. We utilized the derivative of Euclidean distance (Ex), which constrains the data range to (0, 1). Therefore, this provides better

data representation and leads to a better fitting to a normal distribution compared to Euclidean distance. As z-values are mainly used for normal distributions, using Ex values allows us to apply z-scores to represent the deviation to the mean of Ex values. We have clarified this on pg. 45-46.

Sincerely,
Mikhail Savitski

Reviewers' comments:

Reviewer #1 (Remarks to the Author):

Dear authors

Thank you for the clarifications. Many factors, as highlighted by Mikhail Savitski, can affect the T_m or the thermal stability of protein thus posing challenge for mechanistic interpretation and validation of TPP/MS-CETSA data. Thermal proximity co-aggregation (TPCA) is a proposed mechanistic model and an analytical framework for identifying proteins with changes in thermal stability that could arise from modulation of their interactions with other proteins. In this work, you have further demonstrated that it can be used to identify new protein-protein interactions. As TPCA is relatively new concept and may not be obvious mechanistically to most reader why proteins from same protein complex exhibit similarity insolubility, I like to request introducing the term and concept "Thermal Proximity Co-aggregation" in the introduction to put analytical methodology presented later into context. Specifically, we believe that proteins in the same protein complex co-aggregate upon heat denaturation and precipitate out of solution together, thus leading to non-random similarity in insolubility at different denaturing temperature.

1. Based on your clarification, it seems you have assessed the reproducibility of your experiments based on reporter ion intensity. As reporter ion intensity can span few orders of magnitude and since subsequent analysis are performed using normalised adjusted solubility instead, I like to request reassessing the reproducibility of your data based on normalised adjusted solubility of proteins at each temperature point. In addition, the analysis presented in Figure 1d should be performed using normalised adjusted solubility.

2. For figure 6b, are the distribution of TED values between non-translocating and translocating protein groups statistically different?

Reviewer #2 (Remarks to the Author):

The resubmitted manuscript entitled “Temporal dynamics of protein complex formation and dissociation during human cytomegalovirus infection” by Hashimoto et al utilizes thermal proteome profiling (TPP) and cellular thermal shift assays in combination with mass spectrometry quantification (MS-CETSA) to identify viral-protein complexes during the CMV life cycle. The authors have addressed this reviewer’s comments by modifying the text and including additional results. The manuscript utilizes advanced proteome analysis to discover the function of several cellular proteins during the late stage of the CMV life cycle. The data is consistent with the overall conclusion of the manuscript and would advance the understanding of CMV biology.

Reviewer #3 (Remarks to the Author):

I am very happy with how the authors have addressed my comments and recommend publication.

We thank the editor and reviewers for the timely assessment of our manuscript. We were delighted to hear that the reviewers were positive about our revised manuscript, and that the reviewers #2 and #3 recommended publication. We have now addressed the additional minor points raised by reviewer #1, and have added two new supplementary figure panels. Our responses to the reviewer comments are marked with “>**Response:**”.

Reviewer #1 (Remarks to the Author):

Dear authors

Thank you for the clarifications. Many factors, as highlighted by Mikhail Savitski, can affect the T_m or the thermal stability of protein thus posing challenge for mechanistic interpretation and validation of TPP/MS-CETSA data. Thermal proximity co-aggregation (TPCA) is a proposed mechanistic model and an analytical framework for identifying proteins with changes in thermal stability that could arise from modulation of their interactions with other proteins. In this work, you have further demonstrated that it can be used to identify new protein-protein interactions. As TPCA is relatively new concept and may not be obvious mechanistically to most reader why proteins from same protein complex exhibit similarity insolubility, I like to request introducing the term and concept “Thermal Proximity Co-aggregation” in the introduction to put analytical methodology presented later into context. Specifically, we believe that proteins in the same protein complex co-aggregate upon heat denaturation and precipitate out of solution together, thus leading to non-random similarity in insolubility at different denaturing temperature.

>Response: As the reviewer suggests, we now introduce the concept of “thermal proximity coaggregation” in the introduction, along with an expanded description of the principle underlying this technique (pg. 6-7).

1. Based on your clarification, it seems you have assessed the reproducibility of your experiments based on reporter ion intensity. As reporter ion intensity can span few orders of magnitude and since subsequent analysis are performed using normalised adjusted solubility instead, I like to request reassessing the reproducibility of your data based on normalised adjusted solubility of proteins at each temperature point. In addition, the analysis presented in Figure 1d should be performed using normalised adjusted solubility.

>Response: We appreciate the reviewer’s comment. We initially assessed the reproducibility of our data based on reporter ion intensities given that this is a routine and necessary assessment, expected to be shown in papers using TMT quantification. We now have performed the additional analysis suggested by the reviewer. We include scatterplots of log transformed normalized adjusted solubility between replicates at each temperature point (new Supplementary Figure S1e). The data were normalized to the 37 °C point (i.e., first temperature point). As the figure indicates, the normalized adjusted solubilities show high correlation values for the 47-64 °C points. As expected, in this measurement the correlations are slightly less for the two temperature points immediately following 37 °C (40 and 44 °C), given that the range of values is small. Therefore, the reproducibility in our data is also demonstrated by the standard deviation of the distance to the midline of the data points (schematic inset on the left) for each temperature (Std. values on scatterplots). The overall low standard deviation further supports the high reproducibility between replicates. Altogether, these additional analyses agree with our normalization by reporter ion intensity. To be inclusive, we opted to show both of these analyses

in the paper, as the users of TMT analyses would also expect to see the assessment of reporter ion intensities.

> As the reviewer suggested, we also performed the PCA analysis for the normalized adjusted solubility values and have included this data in Supplementary Figure 1d. Based on normalized adjusted solubility values, the PCA analysis clearly separated the 10 different temperatures (first panel) and showed reproducibility between replicates at each temperature. Notably, this type of normalization makes the effect of temperature more visible in PCA space, while masking the contribution of infection time. Our subsequent analyses, based on these normalized adjusted solubilities and Euclidean distances, show that infection time points correlate with differences in co-aggregation profiles, and we could ascribe functions to selected alterations. To highlight the separation of the data by both temperature and infection time points, we show both normalization analyses in our paper.

2. For figure 6b, are the distribution of TED values between non-translocating and translocating protein groups statistically different?

>Response: Yes. As assessed using a Mann-Whitney U-test on the distribution of TED values for these protein groups, the difference between the non-translocating and translocating proteins (excluding ribosomal proteins) was found to be statistically significant. This analysis further supports that TED helps to capture the changes in thermal shift. We have now included the statistical values in the legend of Figure 6b and in Supplementary Figure 10.

Reviewer #2 (Remarks to the Author):

The resubmitted manuscript entitled “Temporal dynamics of protein complex formation and dissociation during human cytomegalovirus infection” by Hashimoto et al utilizes thermal proteome profiling (TPP) and cellular thermal shift assays in combination with mass spectrometry quantification (MS-CETSA) to identify viral-protein complexes during the CMV life cycle. The authors have addressed this reviewer’s comments by modifying the text and including additional results. The manuscript utilizes advanced proteome analysis to discover the function of several cellular proteins during the late stage of the CMV life cycle. The data is consistent with the overall conclusion of the manuscript and would advance the understanding of CMV biology.

>Response: We thank the reviewer for indicating that all the questions were adequately addressed.

Reviewer #3 (Remarks to the Author):

I am very happy with how the authors have addressed my comments and recommend publication.

>Response: We thank the reviewer for all the positive comments.